# Autophagy regulates inflammatory programmed cell death via turnover of RHIM-domain proteins

**Junghyun Lim[1], Hyunjoo Park[2], Jason Heisler[2], Timurs Maculins[1], Merone Roose-Girma[3], Min Xu[2], Brent Mckenzie[2], Menno van Lookeren Campagne[4], Kim Newton[5], Aditya Murthy[1]***

[1]Department of Cancer Immunology, Genentech, South San Francisco, United States; [2]Department of Translational Immunology, Genentech, South San Francisco, United States; [3]Department of Molecular Biology, Genentech, South San Francisco, United States; [4]Department of Immunology, Genentech, South San Francisco, United States; [5]Department of Physiological Chemistry, Genentech, South San Francisco, United States

*For correspondence:
murthy.aditya@gene.com

**Abstract** RIPK1, RIPK3, ZBP1 and TRIF, the four mammalian proteins harboring RIP homotypic interaction motif (RHIM) domains, are key components of inflammatory signaling and programmed cell death. RHIM-domain protein activation is mediated by their oligomerization; however, mechanisms that promote a return to homeostasis remain unknown. Here we show that autophagy is critical for the turnover of all RHIM-domain proteins. Macrophages lacking the autophagy gene *Atg16l1* accumulated highly insoluble forms of RIPK1, RIPK3, TRIF and ZBP1. Defective autophagy enhanced necroptosis by Tumor necrosis factor (TNF) and Toll-like receptor (TLR) ligands. TNF-mediated necroptosis was mediated by RIPK1 kinase activity, whereas TLR3- or TLR4-mediated death was dependent on TRIF and RIPK3. Unexpectedly, combined deletion of *Atg16l1* and *Zbp1* accelerated LPS-mediated necroptosis and sepsis in mice. Thus, ZBP1 drives necroptosis in the absence of the RIPK1-RHIM, but suppresses this process when multiple RHIM-domain containing proteins accumulate. These findings identify autophagy as a central regulator of innate inflammation governed by RHIM-domain proteins.
DOI: https://doi.org/10.7554/eLife.44452.001

## Introduction

Programmed cell death plays a central role in dictating tolerogenic or immuno-stimulatory responses. To leverage these pathways therapeutically, it is critical to understand how immune-suppressive versus inflammatory modes of cell death (e.g. necroptosis and pyroptosis) are regulated. RHIM-domain containing proteins have emerged as central nodes in inflammatory signaling mediated by cytokines or microbial antigens (also known as microbe-associated molecular patterns/MAMPs) (*de Almagro and Vucic, 2015*; *Kajava et al., 2014*; *Pasparakis and Vandenabeele, 2015*). Receptor interacting protein kinase 1 (RIPK1) kinase activity drives caspase 8-dependent apoptosis as well as pro-inflammatory necroptosis dependent on RIPK3 and its substrate mixed lineage kinase domain like (MLKL) (*Cho et al., 2009*; *He et al., 2009*; *Sun et al., 2012*; *Zhang et al., 2009*; *Zhao et al., 2012*). Additionally, the cytosolic adaptor Toll/IL-1 receptor (TIR) domain-containing adaptor protein inducing interferon−β (TRIF) and innate sensor Z-DNA binding protein 1 (ZBP1) can directly interact with RIPK1 and/or RIPK3 via their RHIM domains, thereby stabilizing downstream signaling (*He et al., 2011*; *Kaiser et al., 2013*; *Lin et al., 2016*; *Newton et al., 2016*; *Thapa et al., 2016*). While the role of these proteins in inflammation and cell death have been elucidated via

genetic deletion, mechanisms which drive a return to homeostasis have remained poorly described (*Cuchet-Lourenço et al., 2018*; *Ito et al., 2016*; *Mocarski et al., 2014*; *Ofengeim et al., 2017*; *Weinlich et al., 2017*). RHIM-dependent oligomerization of TRIF, RIPK1, RIPK3 and ZBP1 is required for their function, but this also results in the generation of amyloid-like structures which require regulated turnover to prevent signal amplification (*Kaiser et al., 2013*; *Li et al., 2012*; *Rebsamen et al., 2009*).

Selective autophagy targets the autophagic machinery to specific cargo via autophagy receptors. Forms of selective autophagy have been identified in the turnover of organelles, cytosolic pathogens and protein aggregates, ultimately driving their turnover via lysosomal degradation (*Anding and Baehrecke, 2017*; *Khaminets et al., 2016*; *Levine et al., 2011*; *Mizushima, 2007*). Putative links between autophagy and TNF-mediated necroptosis of epithelial cells have emerged, but contradictory observations in primary versus transformed cells question how autophagy impacts necroptosis in these models (*Goodall et al., 2016*; *Matsuzawa-Ishimoto et al., 2017*). In the current study, we investigated the role of autophagy in innate immunity driven by macrophage activation. We found that autophagy is critical for the turnover of highly insoluble complexes containing TRIF, RIPK1, RIPK3 or ZBP1. Defective autophagy enhanced cytokine production and necroptosis driven by activators of RHIM-domain proteins. Unexpectedly, we observed that ZBP1 dampens necroptosis in a context-specific manner, since deletion of *Zbp1* in an autophagy deficient background exacerbated necroptosis driven by TRIF. Thus, we identify autophagy as an upstream regulator of RHIM-domain proteins and reveal a non-canonical, immunosuppressive function of ZBP1 upon defective autophagy.

## Results

### Defective autophagy enhances RIPK1-dependent and independent forms of macrophage death

We first asked whether TNF or TLR ligands promote cell death in autophagy-deficient macrophages by deleting *Atg16l1*, a core gene in the autophagy pathway. Stimulation with TNF or TLR ligands does not significantly induce death of wild-type macrophages, with the exception of TLR3 which can drive caspase-8 mediated apoptosis via TRIF (*Kaiser et al., 2013*; *Gentle et al., 2017*; *Kawasaki and Kawai, 2014*). However, combining inflammatory stimuli with caspase-inhibitors are established methods to study inflammatory cell death via necroptosis in vitro and in vivo (*de Almagro and Vucic, 2015*; *Pasparakis and Vandenabeele, 2015*; *Weinlich et al., 2017*). *Atg16l1* deficient bone marrow-derived macrophages (*Atg16l1*-cKO BMDMs) did not exhibit increased sensitivity to TNF alone, but they were more sensitive than wild-type (*Atg16l1*-WT) BMDMs to necroptotic stimulus of TNF plus pancaspase inhibitor zVAD-fmk (*Figure 1A*). Blocking the kinase activity of RIPK1 with Necrostatin-1 (Nec-1) reduced death in both *Atg16l1*-WT and *Atg16l1*-cKO BMDMs (*Figure 1A*), consistent with active RIPK1 engaging RIPK3 downstream of TNFR1 (*Newton et al., 2014*). Death induced by TLR2, TLR3, TLR4, TLR7/8 or TLR9 ligands plus zVAD-fmk was also enhanced by *Atg16l1* deficiency (*Figure 1B*; *Figure 1—figure supplement 1A*). While Nec-1 suppressed necroptosis by TLR2, TLR7/8 and TLR9 ligands in both genotypes, it was less effective at preventing TLR3- or TLR4-mediated death in *Atg16l1*-cKO BMDMs compared with *Atg16l1*-WT BMDMs (*Figure 1B*; *Figure 1—figure supplement 1A*). IL-1β release by *Atg16l1*-cKO BMDMs was also elevated upon LPS-mediated necroptosis independent of RIPK1 inhibition (*Figure 1—figure supplement 1B*). The modest effect of Nec-1 in *Atg16l1*-cKO BMDMs may stem from it blocking necroptosis due to autocrine TNF production, which was elevated upon *Atg16l1* deletion (*Figure 1—figure supplement 1C*), but not death due to other mechanisms activating RIPK3.

Core autophagy genes can contribute to autophagy-independent functions in innate immunity (*Codogno et al., 2011*; *Fletcher et al., 2018*; *Heckmann et al., 2017*). Conclusive proof of autophagy in suppressing necroptosis therefore requires a genetic approach assessing multiple autophagy-related genes. We established a non-viral gene editing protocol in primary murine macrophages by comparing CRISPR/Cas9-mediated deletion of enhanced green fluorescent protein (eGFP; schematic in *Figure 1—figure supplement 2A*). Efficient gene knockdown was achieved in monocyte- and bone marrow-derived macrophages (*Figure 1—figure supplement 2B*, eGFP deletion; *Figure 1—figure supplement 2C,D*, *Ptprc*/CD45 deletion). We used this method in wild-type (WT) BMDMs to

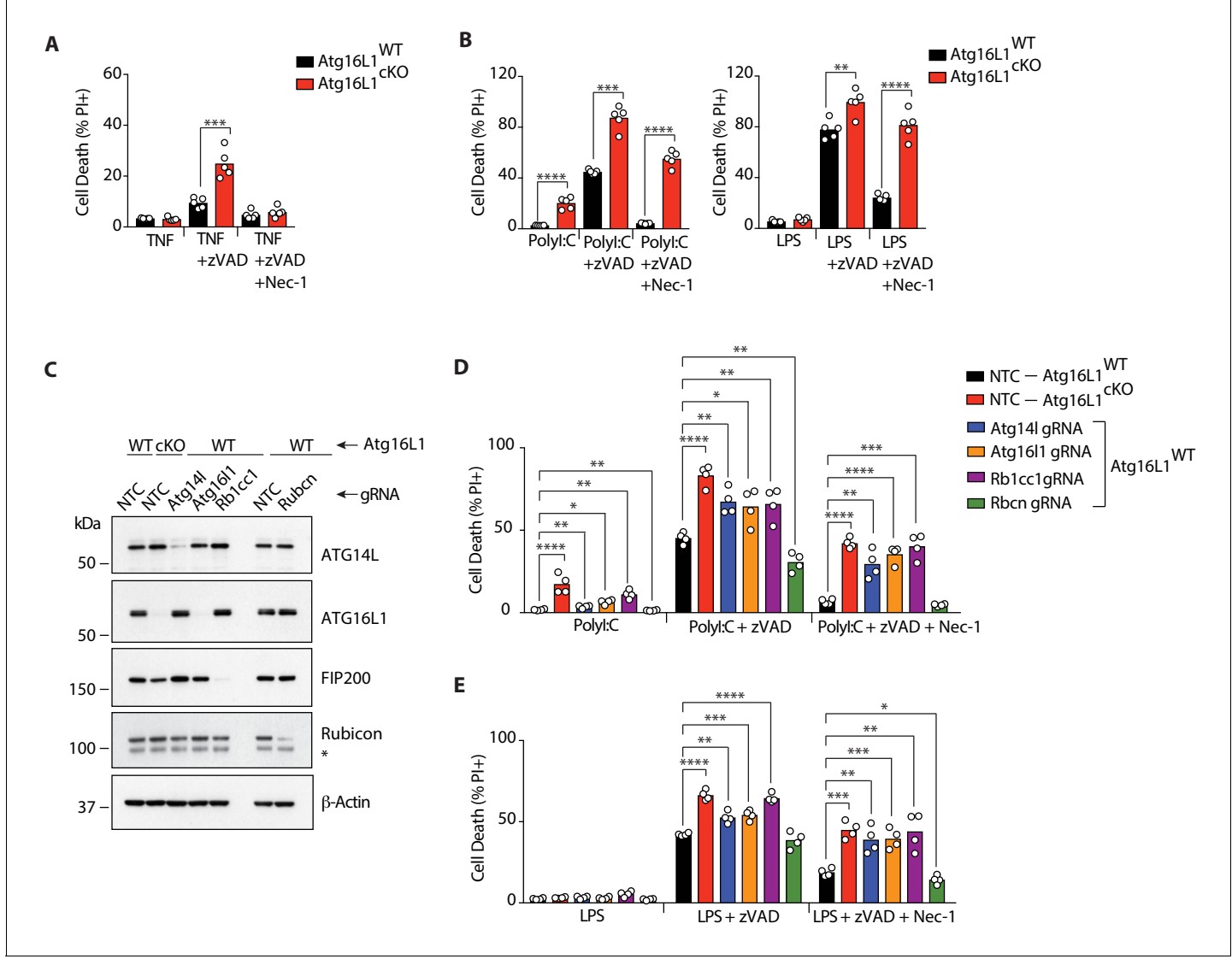

**Figure 1.** Defective autophagy enhances RIPK1-dependent and independent necroptosis. (A, B) Cell death assayed by Propidium Iodide (PI) staining and live-cell imaging for 12–16 hr (n = 5). BMDMs from mice of the indicated genotypes were treated with combinations of TNF/zVAD/Nec-1 (A) or PolyI:C/zVAD/Nec-1 and LPS/zVAD/Nec-1 (B). (C) Immunoblots confirming deletion of autophagy genes in BMDMs of indicated genotypes using RNP electroporation. NTC = non targeting control gRNA. (D, E) Cell death assayed under combinations of PolyI:C/zVAD/Nec-1 (D) or LPS/zVAD/Nec-1 (E) treatment (n = 4). Data in (A, B) are representative of four independent experiments; (C–E) are representative of two independent experiments. *p<0.05, **p<0.01, ***p<0.001, ****p<0.0001. Bar graphs depict mean.

DOI: https://doi.org/10.7554/eLife.44452.002

The following source data and figure supplements are available for figure 1:

**Source data 1.** Defective autophagy enhances RIPK1-dependent and independent necroptosis.
DOI: https://doi.org/10.7554/eLife.44452.005

**Figure supplement 1.** Elevated cell death and cytokine production by *Atg16l1*-cKO BMDMs.
DOI: https://doi.org/10.7554/eLife.44452.003

**Figure supplement 1—source data 1.** Elevated cell death and cytokine production by Atg16l1-cKO BMDMs.
DOI: https://doi.org/10.7554/eLife.44452.006

**Figure supplement 2.** CRISPR-mediated deletion of genes in primary BMDMs.
DOI: https://doi.org/10.7554/eLife.44452.004

**Figure supplement 2—source data 2.** CRISPR-mediated deletion of genes in primary BMDMs.
DOI: https://doi.org/10.7554/eLife.44452.007

delete canonical autophagy genes *Atg14l*, *Rb1cc1* (encoding FIP200) and *Atg16l1*, as well as *Rubcn* (encoding Rubicon), the principal gene associated with LC3-associated phagocytosis (LAP) (*Heckmann et al., 2017*) (*Figure 1C*). Deletion of *Atg14l* and *Rb1cc1* significantly enhanced PolyI:C/zVAD- or LPS/zVAD-induced necroptosis, whereas *Rubcn* deletion resulted in cell death levels comparable to *Atg16l1*-WT controls (*Figure 1D,E*). As with loss of *Atg16l1*, *Atg14l* and *Rb1cc1*-deficient BMDMs maintained elevated levels of cell death upon Nec-1 treatment (*Figure 1D,E*). These results demonstrate that canonical autophagy suppresses TLR3/4-induced necroptosis in BMDMs and uncover a RIPK1 kinase-independent mode of cell death by TLR3/4 activation when autophagy is perturbed.

## TRIF and RIPK1 drive necroptosis in *Atg16l1*-deficient macrophages

To characterize mode of cell death in *Atg16l1*-cKO BMDMs, we performed CRISPR-mediated deletion of necroptosis mediators *Ripk3* and *Mlkl* as well as *Gsdmd* (Gasdermin d), the final executor of pyroptosis (*Kayagaki et al., 2015*; *Sarhan et al., 2018*; *Shi et al., 2015*). Deletion of *Ripk3* and *Mlkl* but not *Gsdmd* prevented LPS/zVAD- and TNF/zVAD-induced death in both *Atg16l1*-WT and *Atg16l1*-cKO BMDMs, confirming that macrophage death was due to necroptosis (*Figure 2A,B*; *Figure 2—figure supplement 1A*). Recently, *Gsdmd*- and caspase-1-independent secondary pyroptosis mediated by *Nlrp3* and *Pycard* (encoding ASC) was described in murine bone-marrow derived dendritic cells (BMDCs) (*Schneider et al., 2017*). However, neither *Nlrp3* nor *Pycard* knockdown prevented RIPK1 kinase-independent cell death in *Atg16l1*-cKO BMDMs (*Figure 2—figure supplement 1B,C*). Therefore, secondary pyroptosis does not appear to contribute to the death of *Atg16l1*-cKO BMDMs treated with LPS/zVAD or PolyI:C/zVAD. As expected, knockdown of *Ticam1* (encoding TRIF) in *Atg16l1*-WT or *Atg16l1*-cKO BMDMs also decreased necroptosis induced by PolyI:C/zVAD or LPS/zVAD, although additional inhibition of RIPK1 provided a more complete rescue of cell death (*Figure 2C–E*). TLR3 only signals via TRIF (reviewed in *Kawasaki and Kawai, 2014*), so the added protection offered by Nec-1 to wild-type cells is consistent with incomplete knockdown of *Ticam1* (*Figure 2C*).

TNF and type I interferons are proposed to license necroptosis in murine macrophages (*Sarhan et al., 2019*; *Siegmund et al., 2016*), so we tested whether pharmacological blockade of these cytokines would rescue necroptosis in *Atg16l1*-deficient BMDMs. Cells were pre-treated for 36 hr with a control antibody (anti-Ragweed), TNFR2-Fc to block TNF, or anti-IFNAR1 to block Interferon-α Receptor 1, and then a necroptosis stimulus was applied. TNFR2-Fc attenuated necroptosis of *Atg16l1*-WT BMDMs comparably to Nec-1, especially with TLR2 or TLR9 ligands or TNF itself (*Figure 2—figure supplement 2A,E,F*), but also with TLR3, TLR4 or TLR7 ligands (*Figure 2—figure supplement 2B,C,D*). In contrast, TNFR2-Fc failed to inhibit necroptosis in *Atg16l1*-cKO BMDMs treated with TLR2, TLR7 or TLR9 ligands (*Figure 2—figure supplement 2A,D,E*), despite inhibiting necroptosis induced by TLR3 or TLR4 ligands or TNF comparably to Nec-1 (TNFR2-Fc, *Figure 2—figure supplement 2B, C, F*). Thus, while TNF contributes to enhanced necroptosis of *Atg16l1*-deficient BMDMs, activation of RIPK1 contributes more to this phenotype, perhaps indicating the involvement of multiple death receptors.

Consistent with the results of *McComb et al. (2014)*, pharmacological blockade of IFNAR1 prevented TLR2, TLR3, TLR4, or TLR9-induced necroptosis in *Atg16l1*-WT BMDMs (*Figure 2—figure supplement 2A,B,C,E*). IFNAR1 blockade also decreased TLR3- or TLR4-induced necroptosis of *Atg16l1*-deficient BMDMs more than Nec-1 (*Figure 2—figure supplement 2B,C*). However, it only provided modest protection to *Atg16l1*-cKO BMDMs treated with TLR2, TLR7, or TLR9 ligands or TNF (*Figure 2—figure supplement 2A,D,E,F*). Accordingly, phosphorylation of STAT1, which is associated with Type I interferon signaling, was increased in *Atg16l1*-cKO BMDMs following LPS/zVAD treatment (*Figure 2—figure supplement 2G*). These data indicate that signaling by Type I interferons contributes to enhanced necroptosis of *Atg16l1*-deficient BMDMs, particularly with ligands that activate TRIF.

## Loss of autophagy results in accumulation of active forms of TRIF, RIPK1 and RIPK3 during necroptosis

We asked whether loss of *Atg16l1* caused TRIF, RIPK1, or RIPK3 to accumulate in necroptotic BMDMs, as this would provide direct biochemical evidence that autophagy promotes their turnover.

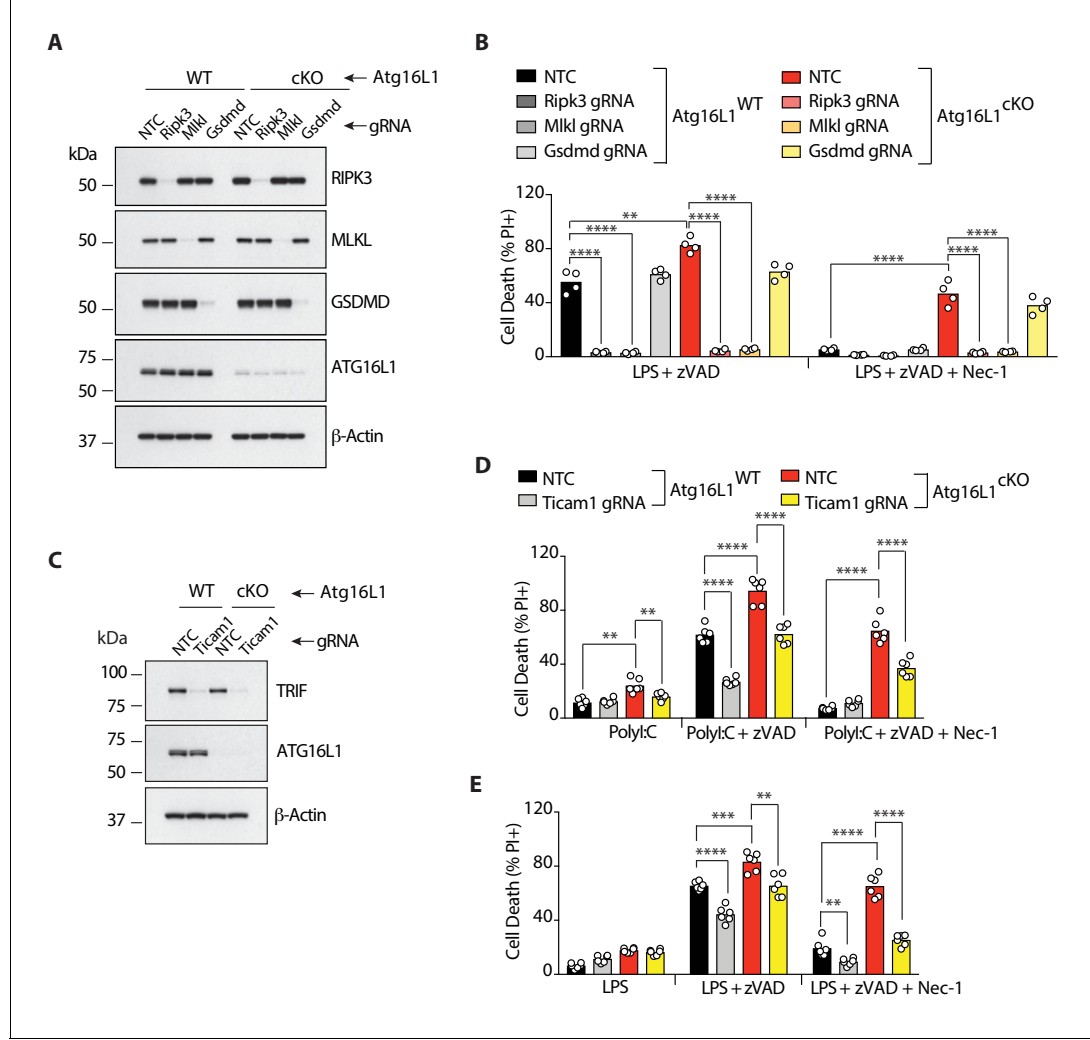

**Figure 2.** RIPK3, MLKL and TRIF are required for RIPK1-independent necroptosis in *Atg16l1*-deficient BMDMs. (A-E) Immunoblot (**A, C**) and cell death assays (**B, D, E**) of BMDMs from mice of indicated genotypes treated with combinations of LPS/zVAD/Nec-1 following CRISPR-mediated deletion of RIPK3, MLKL or GSDMD (**A, B**) (n = 4) or TRIF (**C–E**) (n = 6). Cell death assayed by PI staining and live-cell imaging for 12–16 hr. Data in (**A, B**) are representative of three independent experiments; (**C, D, E**) are representative of four independent experiments. **p<0.01, ***p<0.001, ****p<0.0001. Bar graphs depict mean. NTC = non targeting gRNA.

DOI: https://doi.org/10.7554/eLife.44452.008

The following source data and figure supplements are available for figure 2:

**Source data 1.** RIPK3, MLKL and TRIF are required for RIPK1-independent necroptosis in Atg16l1-deficient BMDMs.
DOI: https://doi.org/10.7554/eLife.44452.010

**Figure supplement 1.** RIPK3 and MLKL drive RIPK1-dependent, TNF-mediated necroptosis; TRIF drives RIPK1-independent, PolyI:C-mediated necroptosis in *Atg16l1*-cKO BMDMs.
DOI: https://doi.org/10.7554/eLife.44452.009

**Figure supplement 1—source data 1.** RIPK3 and MLKL drive RIPK1-dependent, TNF-mediated necroptosis; TRIF drives RIPK1-independent, PolyI:C mediated necroptosis in Atg16l1-cKO BMDMs.
DOI: https://doi.org/10.7554/eLife.44452.011

**Figure supplement 2.** TNF and Type I interferon license necroptosis in BMDMs.
DOI: https://doi.org/10.7554/eLife.44452.013

**Figure supplement 2—source data 2.** TNF and Type I interferon license necroptosis in BMDMs.
DOI: https://doi.org/10.7554/eLife.44452.012

TRIF appeared to be monomeric in untreated BMDMs of both genotypes, running as a single band in the detergent (NP-40) soluble fraction below the 100-kDa mark in an SDS-PAGE gel (*Figure 3A, B*). LPS was sufficient to elicit a smear of NP40-insoluble, slower migrating TRIF species that was far more prominent in *Atg16l1*-cKO BMDMs than in *Atg16l1*-WT BMDMs (*Figure 3B*). CRISPR-knock-down of *Ticam1* (encoding TRIF) in *Atg16l1*-cKO cells confirmed that the high-MW species were TRIF (*Figure 3B*, *Atg16l1*-cKO, *Ticam1*-gRNA sample). High-MW species of autophosphorylated RIPK1 (p-RIPK1, Ser166/Thr169) as well as total RIPK1 largely accumulated in the NP-40 insoluble fraction after treatment with LPS plus zVAD; these were more abundant in *Atg16l1*-cKO BMDMs than in *Atg16l1*-WT BMDMs (*Figure 3C,D*). Slower migrating species of both autophosphorylated RIPK3 (Thr231/Ser232) and total RIPK3 in the insoluble fraction after treatment with LPS plus zVAD were also elevated in *Atg16l1*-cKO BMDMs when compared with *Atg16l1*-WT BMDMs (*Figure 3E, F*). Knockdown of *Ticam1* decreased the amount of autophosphorylated RIPK3 and autophosphorylated RIPK1 in *Atg16l1*-cKO cells (*Figure 3D,F*). Loss of *Atg16l1* did not appear to affect the abundance of monomeric TRIF, RIPK1, or RIPK3 in the detergent soluble fraction (*Figure 3A,C,E*), suggesting that autophagic turnover specifically regulates levels of activated/modified forms of TRIF, RIPK1 and RIPK3. Consistent with reduced autophagic turnover of RHIM proteins enhancing TLR4-induced necroptosis, *Atg16l1*-cKO BMDMs treated with LPS plus zVAD contained more MLKL phosphorylated at Ser345, a marker of necroptosis (*Sun et al., 2012*; *Cai et al., 2014*; *Chen et al., 2014*; *Dondelinger et al., 2014*; *Wang et al., 2014*), when compared to their wild-type counterparts (*Figure 4—figure supplement 1A*).

Measuring the kinetics of cell death revealed that LPS/zVAD-induced necroptosis killed more than 80% of *Atg16l1*-cKO BMDMs within 3 hr of treatment, compared to approximately 50% of *Atg16l1*-WT BMDMs (*Figure 4A*, *Figure 4—figure supplement 1B*). Interestingly, *Atg16l1*-deficient BMDMs were more sensitive than *Atg16l1*-WT BMDMs to TLR3 engagement alone without zVAD (*Figure 4—figure supplement 1C*), consistent with TLR3 and TRIF having the capacity to assemble a death-inducing signaling complex that activates caspase-8 (*Zinngrebe et al., 2016*). We analyzed TRIF, RIPK1 and RIPK3 in the detergent-insoluble fraction in the first 6 hr after treatment with LPS/zVAD, and found that high-MW forms of TRIF accumulated transiently and with similar kinetics in both genotypes, peaking at 1 hr, and to a greater extent in the absence of *Atg16l1* (*Figure 4B*). Accumulation of autophosphorylated, high-MW RIPK1 and RIPK3 peaked at 2 hr after LPS/zVAD treatment in both genotypes, with greater accumulation in *Atg16l1*-cKO BMDMs (*Figure 4C,D*). Autophosphorylated RIPK1 and RIPK3 were also ubiquitinated with Met1- or Lys63-linked chains, with greater abundance in *Atg16l1*-cKO BMDMs (*Figure 4E*). Indeed, accumulation of ubiquitinated protein aggregates is a hallmark of defective autophagy (*Kwon and Ciechanover, 2017*; *Dikic, 2017*).

To confirm that accumulation of modified forms of TRIF, RIPK1 and RIPK3 occurred due to defective lysosomal turnover via autophagy, pharmacological inhibition of lysosomal function was performed in WT BMDMs during necroptosis. Consistent with our genetic models, treatment of WT BMDMs with Bafilomycin A1, an inhibitor lysosomal vacuolar H-ATPases, resulted in accumulation of high-MW forms of TRIF and RIPK1 in detergent insoluble fractions during LPS/zVAD-mediated necroptosis over 6 hr (*Figure 4—figure supplement 2A–C*). In contrast, basal turnover of low-MW TRIF, RIPK1, and RIPK3 was not perturbed in a reproducible manner by Bafilomycin A1. For comparison, inhibition of proteasomal degradation with MG-132 caused a very subtle increase in low-MW TRIF and RIPK1, whereas RIPK3 appeared unaffected *Figure 4—figure supplement 2D–F*). Collectively, these data indicate that lysosomal turnover via autophagy is critical for preventing the accumulation of active TRIF, RIPK1 and RIPK3, and its loss exacerbates necroptotic signaling.

## The autophagy receptor TAX1BP1 prevents TRIF-mediated necroptosis

Having demonstrated a role for core autophagy genes in macrophage necroptosis (*Figure 1*), we analyzed autophagic flux in WT BMDMs during LPS-mediated necroptosis. LC3, a critical component of the mature autophagosome membrane that receives autophagic cargo, is lipidated during the process of autophagy. Additionally, selective autophagy receptors which can potentially associate with cytosolic substrates are trafficked to autophagosomes as a consequence of autophagic flux (*Anding and Baehrecke, 2017*; *Dikic, 2017*). Treatment with LPS/zVAD in the presence of Bafilomycin A1 to halt autophagic flux revealed accumulation of lipidated LC3B (LC3-II) in WT BMDMs (*Figure 4—figure supplement 2G*). Levels of the autophagy receptors SQSTM1/p62, TAX1BP1 and

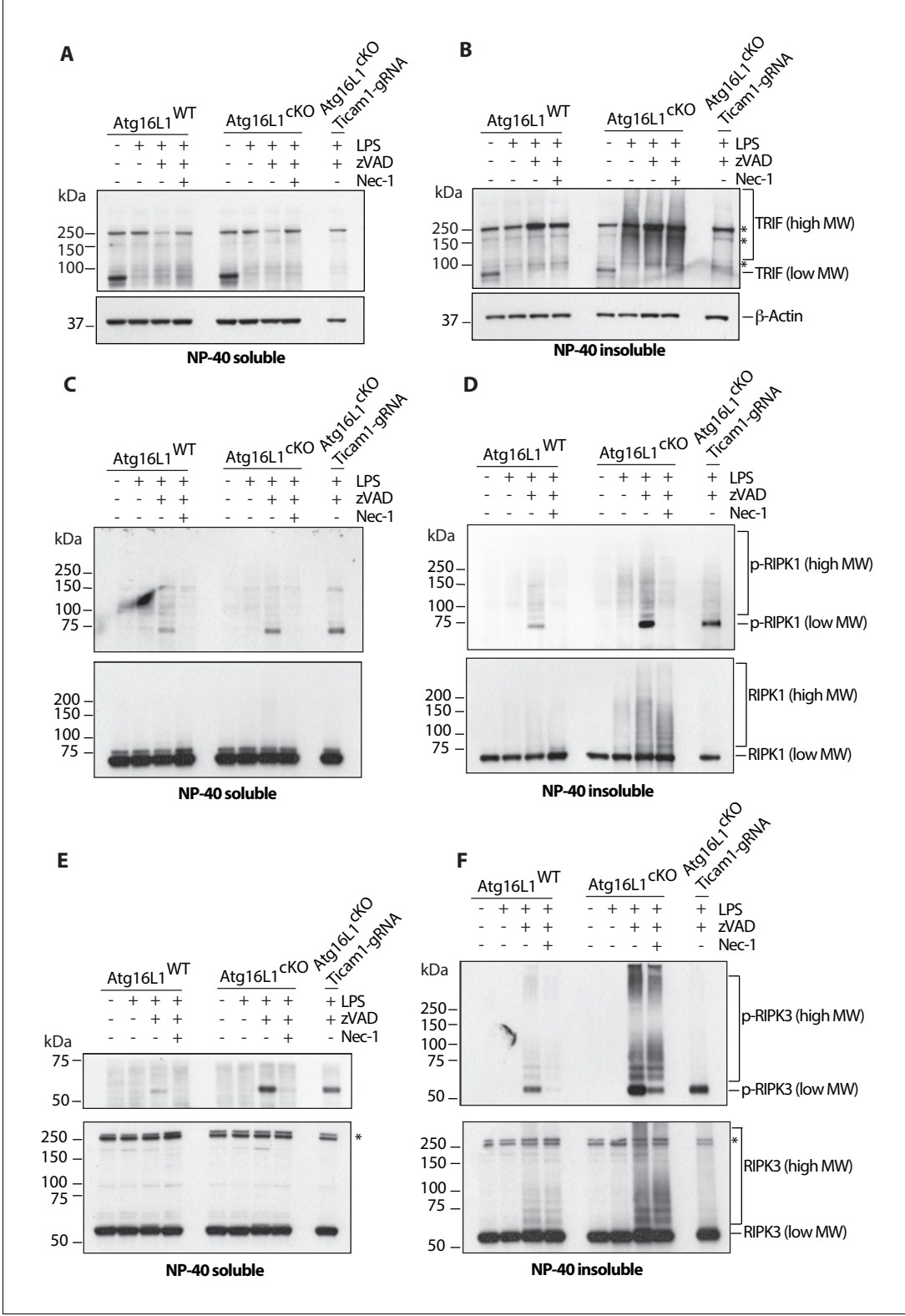

**Figure 3.** Loss of *Atg16l1* drives accumulation of detergent insoluble, high molecular weight TRIF, RIPK1, RIPK3 and enhances RIPK1/RIPK3 phosphorylation. (**A, B**) Immunoblots of TRIF in *Atg16l1*-WT and *Atg16l1*-cKO BMDM lysates following 4 hr of treatment with indicated combinations of LPS/zVAD/Nec-1 and enrichment of NP-40 soluble (**A**) or insoluble (**B**) fractions. (**C, D**) immunoblots for autophosphorylated RIPK1 (Ser166/Thr169, p-RIPK1) and total RIPK1 in *Atg16l1*-WT and *Atg16l1*-cKO BMDM lysates following 4 hr of treatment with indicated

*Figure 3 continued on next page*

*Figure 3 continued*

combinations LPS/zVAD/Nec-1 and enrichment of NP-40 soluble (C) or insoluble (D) fractions. (E, F) immunoblot assay for autophosphorylated RIPK3 (Thr231/Ser232, p-RIPK3) and total RIPK3 in *Atg16l1*-WT and *Atg16l1*-cKO BMDM lysates following 4 hr of treatment with indicated combinations of LPS/zVAD/Nec-1 and enrichment of NP-40 soluble (E) or insoluble (F) fractions. Representative data shown from three independent experiments. In all immunoblots, CRISPR-mediated TRIF deletion was performed in *Atg16l1*-cKO BMDMs followed by LPS/zVAD treatment as a negative control. *=non specific bands (n.s.).

DOI: https://doi.org/10.7554/eLife.44452.014

CALCOCO1 were elevated upon Bafilomycin A1 treatment, especially in the detergent-insoluble fraction (*Figure 4—figure supplement 2G,H*). These data suggest that autophagic flux was ongoing during necroptosis and that autophagy receptors accumulated in the same subcellular compartment as active TRIF, RIPK1 and RIPK3. We and others have observed that the selective autophagy receptor TAX1BP1 suppresses TRIF abundance and TRIF-dependent IFNβ production in BMDMs (*Samie et al., 2018*; *Yang et al., 2017*). These findings are consistent with our unbiased proteomics-based identification of SQSTM1/p62, TAX1BP1 and CALCOCO1 as candidate selective autophagy receptors in BMDMs during TLR4 activation (*Samie et al., 2018*). Immunoblotting revealed accumulation of SQSTM1/p62, TAX1BP1 and CALCOCO1 with varying kinetics after treatment with LPS/zVAD, and loss of *Atg16l1* further increased the levels of these receptors in the detergent-insoluble fraction (*Figure 5A,B*). We utilized CRISPR-mediated knockdown of these autophagy receptors in WT BMDMs to assess their role in necroptosis induced by ligands that engage TRIF (*Figure 5C*). Loss of *Tax1bp1* enhanced BMDM death by either PolyI:C/zVAD or LPS/zVAD treatment. Necroptosis in both settings was only partially blocked by Nec-1. In contrast, knockdown of *Sqstm1* or *Calcoco1* did not increase TLR3- or TLR4-induced necroptosis in BMDMs (*Figure 5D*). Thus, the autophagy receptor TAX1BP1 suppresses BMDM necroptosis downstream of TRIF.

## Accumulation of ZBP1 protects against necroptosis in *Atg16l1*-deficient macrophages

We recently identified the RHIM-domain containing protein ZBP1 as one of the most highly accumulated proteins in *Atg16l1*-deficient macrophages (*Samie et al., 2018*). Loss of *Atg16l1* resulted in basal accumulation of ZBP1 as shown by a cycloheximide-chase assay (*Figure 6A*). In contrast to TRIF, RIPK1 and RIPK3, basal turnover of ZBP1 was attenuated by lysosomal inhibition, because treatment with Bafilomycin A1, but not MG132, resulted in ZBP1 accumulation in WT BMDMs (*Figure 6—figure supplement 1A*). ZBP1 also accumulated during LPS/zVAD-induced necroptosis, and this was enhanced by *Atg16l1* deletion (*Figure 6—figure supplement 1B*). No high-MW forms of ZBP1 were detected using currently available reagents. The role of the Zα1/Zα2- or RHIM-domains of ZBP1 in its accumulation will need to be addressed in future studies.

ZBP1 has been shown to promote cell death upon accumulation of endogenous or viral nucleic acids (*Thapa et al., 2016*; *Kesavardhana et al., 2017*; *Kuriakose et al., 2016*; *Maelfait et al., 2017*), or genetic deletion of RIPK1-RHIM (*Lin et al., 2016*; *Newton et al., 2016*). Thus, we hypothesized that elevated ZBP1 might engage RIPK3 and contribute to enhanced necroptosis in *Atg16l1*-deficient BMDMs. However, CRISPR-mediated deletion of *Zbp1* in *Atg16l1*-cKO BMDMs (*Figure 6B*) further enhanced LPS/zVAD-induced necroptosis, and this death was prevented by deletion of *Ticam1* but not by Nec-1 (*Figure 6C*; *Figure 6—figure supplement 1C*). Therefore, contrary to expectations, ZBP1 appears to suppress TRIF-mediated necroptosis (*Figure 6C*). Notably, loss of *Zbp1* did not impact the death of autophagy-sufficient BMDMs, suggesting that levels of ZBP1 must cross a certain threshold before suppressing TRIF-dependent necroptosis. To more thoroughly characterize the sensitization conferred by *Zbp1* loss, we measured cell death after performing a dose-titration of LPS in the presence of 20 μM zVAD. CRISPR-mediated deletion of *Zbp1* alone did not impact cell death at any dose of LPS (*Figure 6D*), but combined with defective autophagy, *Zbp1* deletion sensitized cells to necroptosis at low doses of LPS, even in the presence of Nec-1 (*Figure 6E*). Thus, overabundant ZBP1 can antagonize TRIF-mediated necroptosis in *Atg16l1*-deficient macrophages. We confirmed our CRISPR-based observations by generating conditional-knockout mice lacking *Atg16l1* (*Atg16l1*-cKO; *Zbp1*-WT) or *Atg16l1* and *Zbp1* (*Atg16l1*-cKO; *Zbp1*-cKO) in myeloid cells. Consistent with our earlier results, *Zbp1* deletion accelerated LPS/zVAD-mediated

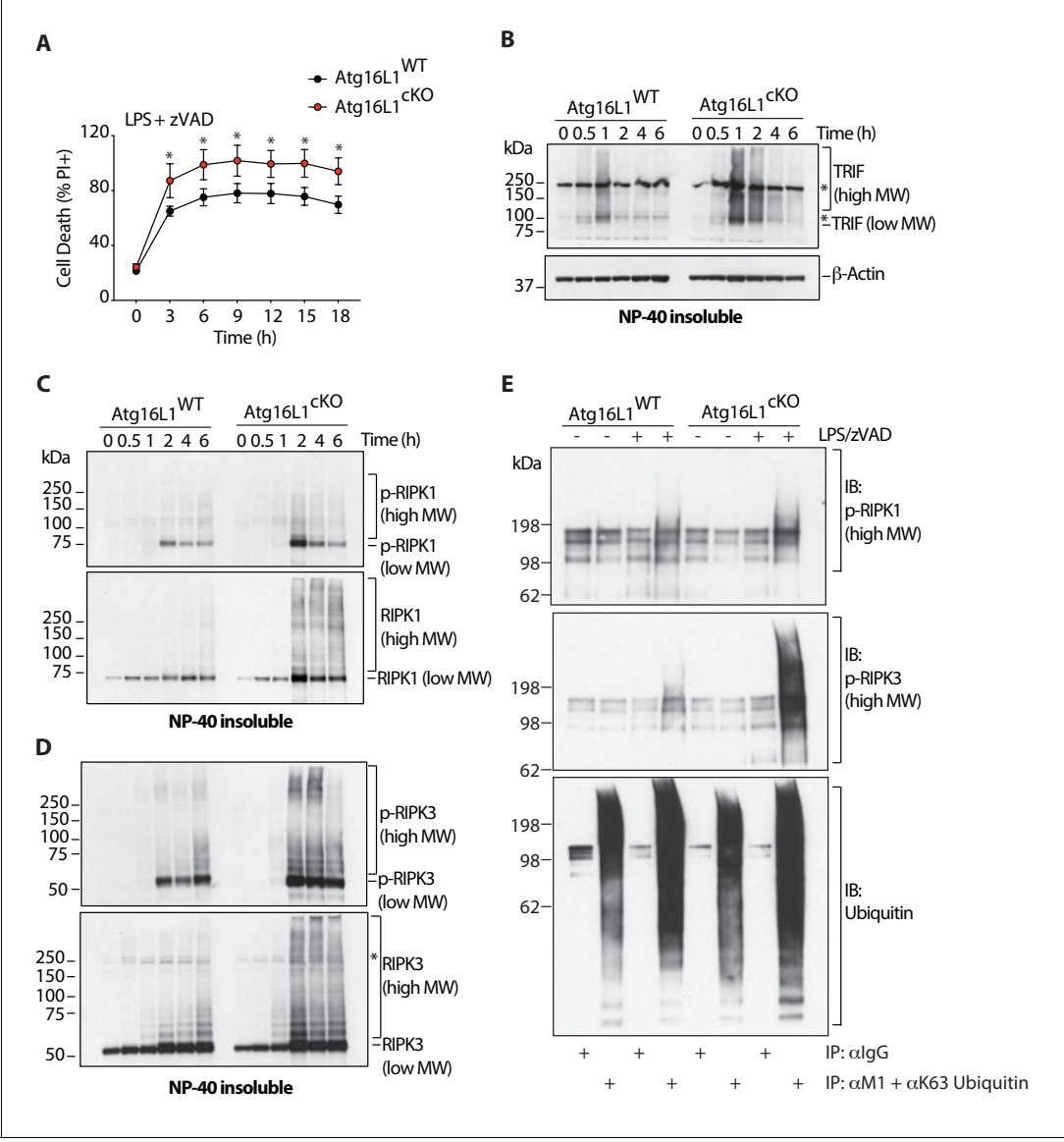

**Figure 4.** Overabundance of TRIF, phosphorylated and ubiquitinated RIPK1 and RIPK3 coincides with accelerated necroptosis of *Atg16l1* deficient BMDMs. (A) Kinetic measurement of cell death over 18 hr of LPS/zVAD treatment (n = 5). (B) Immunoblot of TRIF in NP-40 insoluble fractions of BMDM lysates over 6 hr of LPS/zVAD treatment. (C, D) Immunoblots of autophosphorylated and total RIPK1 (C), RIPK3 (D) in NP-40 insoluble fractions of BMDM lysates treated as in (B). (E) Immunoblots of autophosphorylated RIPK1, RIPK3 and ubiquitin in BMDM lysates following immunoprecipitation of M1 or K63-ubiquitinated proteins after 4 hr of LPS/zVAD treatment. Data in (A) are representative of four independent experiments; (B–D) are representative of three independent experiments; (E) are representative of three independent experiments. *=P < 0.05.

DOI: https://doi.org/10.7554/eLife.44452.015

The following source data and figure supplements are available for figure 4:

**Source data 1.** Overabundance of TRIF, phosphorylated and ubiquitinated RIPK1 and RIPK3 coincides with accelerated necroptosis of ATG16L deficient BMDMs.

DOI: https://doi.org/10.7554/eLife.44452.018

**Figure supplement 1.** Enhanced MLKL activation and accelerated cell death in *Atg16l1* deficient BMDMs following LPS- or PolyI:C-mediated necroptosis.

DOI: https://doi.org/10.7554/eLife.44452.016

**Figure supplement 1—source data 1.** Enhanced MLKL activation and accelerated cell death in ATG16L1 deficient BMDMs following LPS- or PolyI:C-mediated necroptosis.

DOI: https://doi.org/10.7554/eLife.44452.019

**Figure supplement 2.** Lysosomal function and autophagic flux drive turnover of active TRIF, RIPK1 and RIPK3 during necroptosis.

*Figure 4 continued on next page*

*Figure 4 continued*

DOI: https://doi.org/10.7554/eLife.44452.017

**Figure supplement 2—source data 2.** Lysosomal function and autophagic flux drive turnover of active TRIF, RIPK1 and RIPK3 during necroptosis.
DOI: https://doi.org/10.7554/eLife.44452.020

necroptosis of *Atg16l1*-deficient macrophages (*Figure 6—figure supplement 1D*). ZBP1 deficiency did not affect the accumulation of TRIF in the detergent insoluble fraction of *Atg16l1*-cKO BMDMs treated with LPS/zVAD (*Figure 6F*), arguing that ZBP1 interferes with signaling events further downstream. Indeed, compared to *Atg16l1*-cKO BMBMs, BMDMs lacking both *Atg16l1* and *Zbp1* contained more high MW species of autophosphorylated RIPK1 (*Figure 6G*) and autophosphorylated RIPK3 (*Figure 6H*) after treatment with LPS/zVAD.

## Combined deletion of *Atg16l1* and *Zbp1* in myeloid cells accelerates LPS-mediated sepsis

Beyond cellular necroptosis, myeloid-specific loss of *Atg16l1* sensitizes mice to LPS-mediated sepsis in vivo (*Samie et al., 2018*). Since *Zbp1* deletion enhanced TRIF-mediated necroptosis in *Atg16l1*-cKO BMDMs *ex vivo*, we asked whether loss of *Atg16l1* and *Zbp1* in myeloid cells would further exacerbate LPS-mediated sepsis. Intraperitoneal administration of LPS (10 mg per kg body weight) reproduced the previously observed sensitization of *Atg16l1*-cKO mice. Combined loss of *Atg16l1* and *Zbp1* significantly accelerated morbidity, with double-knockout mice succumbing to LPS-driven mortality by 14 hr. Loss of *Zbp1* alone did not impact morbidity (*Figure 7A*; *Figure 7—figure supplement 1A*). *Atg16l1* deficiency in myeloid cells exacerbated LPS-induced IL-1β and TNF in the serum, and the amount of IL-1β was yet higher in *Atg16l1*-cKO; *Zbp1*-cKO double-knockout mice (*Figure 7B*). Together, these results demonstrate that: 1) autophagy regulates ZBP1 abundance in macrophages, 2) elevated ZBP1 suppresses necroptosis when autophagy is perturbed, and 3) loss of both *Atg16l1* and *Zbp1* in myeloid cells accelerates LPS-mediated inflammation, enhancing morbidity *in vivo*.

## Discussion

Macrophages represent primary cellular sensors of the innate immune system. During an inflammatory response, these phagocytes are armed to either propagate or resolve inflammation via antigen uptake and presentation, cytokine production, or induction of programmed cell death. Here, we show that defective autophagy in macrophages leads to an accumulation of modified RIPK1, RIPK3, and TRIF in response to pro-inflammatory signals, precipitating inflammation and necroptosis in a stimulus-dependent manner (*Figure 7—figure supplement 1B,C*). While RIPK1, RIPK3 and TRIF are well-established signaling factors required for necroptosis, our understanding of ZBP1 in this context is less developed. Recent studies have expanded on cytosolic nucleic-acid sensing by ZBP1 (*Thapa et al., 2016*; *Kesavardhana et al., 2017*; *Kuriakose et al., 2016*; *Maelfait et al., 2017*; *Guo et al., 2018*). ZBP1 was also shown to drive RIPK3-mediated necroptosis when the RIPK1-RHIM domain was mutated in vivo (*Lin et al., 2016*; *Newton et al., 2016*). However, its role in other forms of innate signaling is not known. Using an optimized protocol for CRISPR-mediated gene editing in primary macrophages, we revealed context-specific roles of TRIF and ZBP1 in regulating necroptosis and inflammatory cytokine production (*Figure 7—figure supplement 1C*). Specifically, we noted that TRIF can promote RIPK1 kinase-independent, RIPK3-dependent necroptosis when its accumulation is not checked by autophagy. TRIF can engage RIPK3 directly without the need for RIPK1 (*Kaiser et al., 2013*; *Newton et al., 2016*), so a scaffolding role for RIPK1 is unlikely. This notion cannot be confirmed genetically because RIPK1-deficiency alone triggers macrophage death (*Newton et al., 2016*). ZBP1 contains multiple RHIM-domains that support formation of amyloid-like structures (*Li et al., 2012*; *Rebsamen et al., 2009*). Accumulation of ZBP1 induced by defective autophagy may perturb optimal TRIF- or RIPK3- signaling via RHIM-mediated interference. Conclusive evidence for this possibility requires the mutation of ZBP1-RHIM domains in autophagy-deficient cells.

In mammalian cells, RHIM-dependent stacking of RIPK1 and RIPK3 has been described as a key feature of the necrosome (*Li et al., 2012*). RHIM-dependent accumulation of TRIF is also

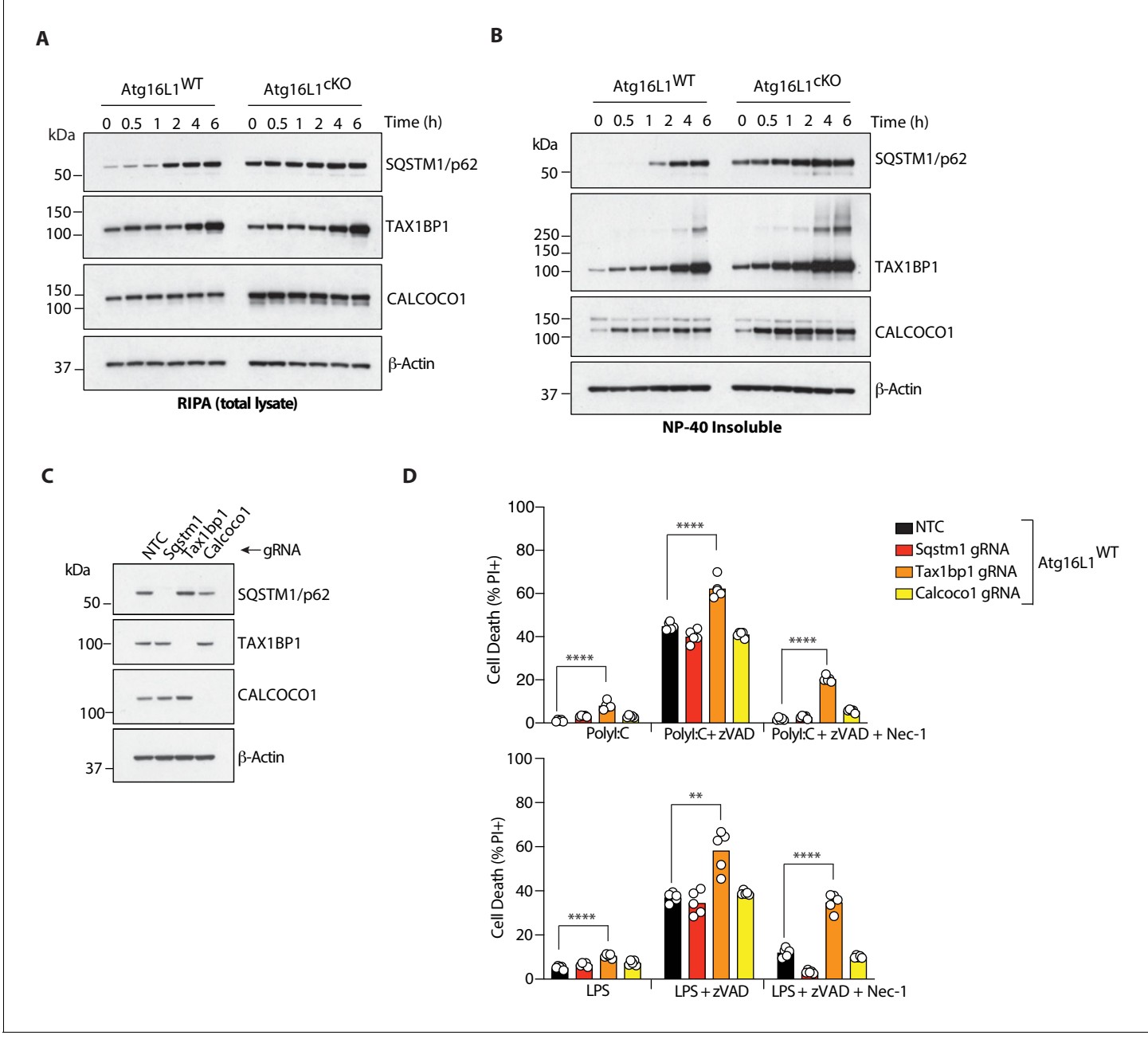

**Figure 5.** The autophagy receptor TAX1BP1 protects against necroptosis by TLR3 or TLR4 ligands. (A, B) Immunoblots of indicated autophagy receptors in total (A) or NP-40 insoluble fractions (B) of BMDM lysates over 6 hr of LPS/zVAD treatment. (C) Immunoblots confirming CRISPR-mediated deletion of indicated autophagy receptor genes in wild-type BMDMs. (D) Cell death assayed by PI staining and live-cell imaging for 12–16 hr following treatment with indicated ligands. Data in (A, B) are representative of three independent experiment; (C, D) are representative of four independent experiments. **p<0.01, ****p<0.0001. Bar graphs depict mean. NTC = non targeting control gRNA.

DOI: https://doi.org/10.7554/eLife.44452.021
The following source data is available for figure 5:

**Source data 1.** The autophagy receptor TAX1BP1 protects against necroptosis by TLR3 or TLR4 ligands.
DOI: https://doi.org/10.7554/eLife.44452.022

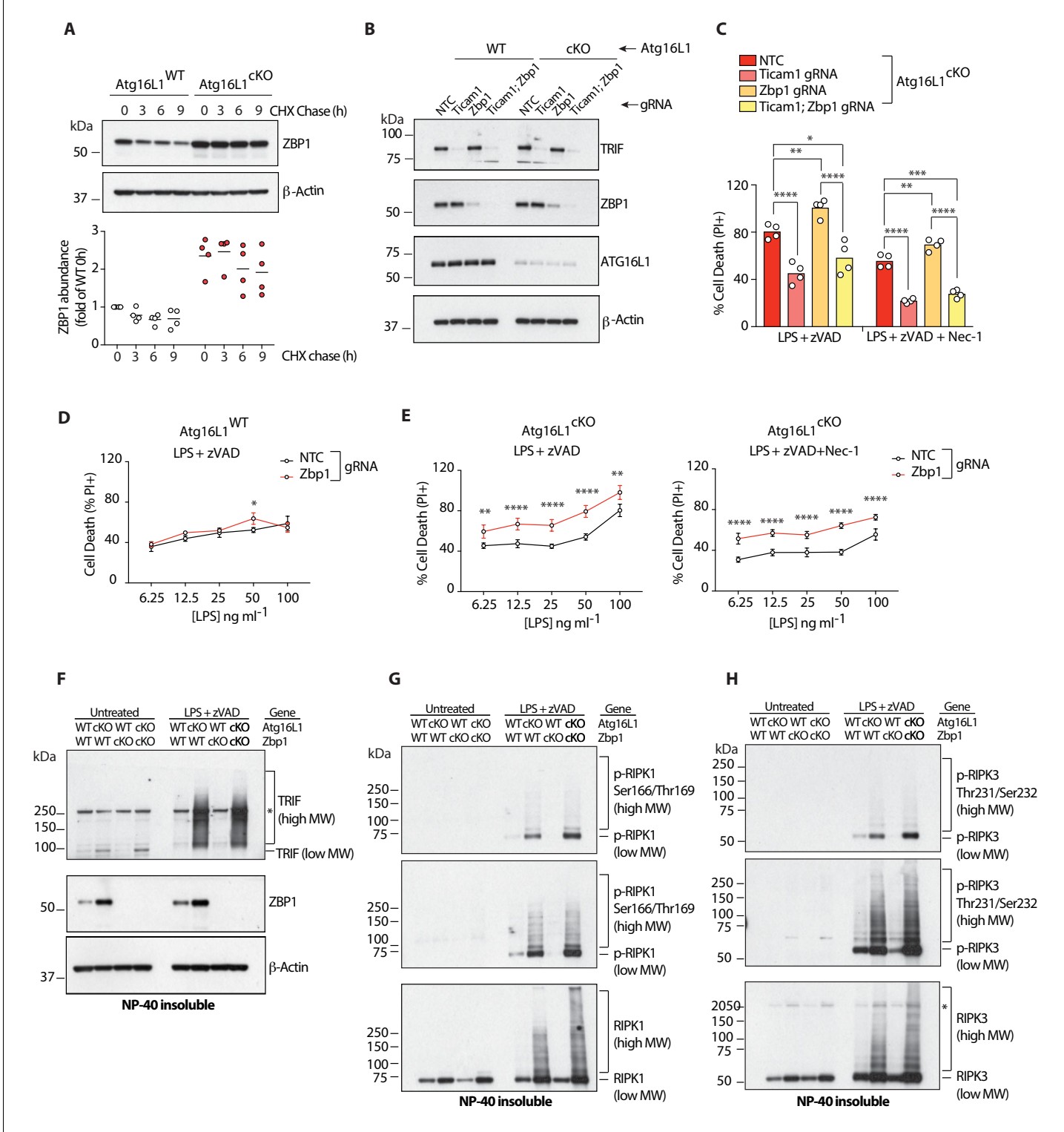

**Figure 6.** Elevated ZBP1 in *Atg16l1*-deficient BMDMs suppresses TRIF-mediated necroptosis. (**A**) ZBP1 turnover in *Atg16l1*-WT and *Atg16l1*-cKO BMDMs following cycloheximide (CHX) treatment for indicated time points. Representative immunoblot (top), ZBP1 quantification by densitometry (bottom) normalized to ZBP1 band intensity in WT samples at 0 hr. (**B, C**) immunoblot (**B**) and cell death (**C**) assays of BMDMs from mice of indicated genotypes treated with combinations of LPS/zVAD/Nec-1 following CRISPR-mediated deletion of *Zbp1*, *Ticam1* or both (n = 4). (**D, E**) cell death assayed in *Atg16l1*-WT or *Atg16l1*-cKO BMDMs following CRISPR-mediated *Zbp1* deletion and a dose titration of LPS in the presence of 20 μM zVAD and/or 30 μM Nec-1 (n = 4). Dot-plots depict mean ±S.D. (**F–H**) immunoblots depicting accumulation of TRIF (**F**), autophosphorylated and total RIPK1
*Figure 6 continued on next page*

*Figure 6 continued*

(**G**), autophosphorylated and total RIPK3 (**H**) in NP-40 insoluble lysates of BMDMs lacking both *Atg16l1* and *Zbp1* following induction of necroptosis via LPS/zVAD for 3 hr. Top panels represent short exposures; middle panels represent long exposures. *=non specific band. Data (**A**) are representative of four independent experiments, densitometry is pooled from four independent experiments. Data in (**B, C**) are representative of three independent experiments; (**D–H**) are representative of two independent experiments. *p<0.05, **p<0.01, ***p<0.001, ****p<0.0001. Bar graphs depict mean. NTC = non targeting control gRNA.

DOI: https://doi.org/10.7554/eLife.44452.023

The following source data and figure supplements are available for figure 6:

**Source data 1.** Elevated ZBP1 in ATG16L1 deficient BMDMs suppresses TRIF-mediated necroptosis.

DOI: https://doi.org/10.7554/eLife.44452.025

**Figure supplement 1.** Loss of *Atg16l1* leads to ZBP1 accumulation; deletion of *Zbp1* in *Atg16l1*-cKO BMDMs enhances TRIF-mediated necroptosis and RIPK3 activation.

DOI: https://doi.org/10.7554/eLife.44452.024

**Figure supplement 1—source data 1.** Loss of Atg16l1 leads to ZBP1 accumulation; deletion of Zbp1 in Atg16l1-cKO BMDMs enhances TRIF-mediated necroptosis and RIPK3 activation.

DOI: https://doi.org/10.7554/eLife.44452.026

acknowledged as a pre-requisite for TRIF-mediated inflammatory signaling (*Gentle et al., 2017*). Biophysical characteristics of RHIM-domain protein complexes include a highly insoluble nature and resistance to denaturation/degradation (*Fowler et al., 2007*; *Kleino et al., 2017*; *Lamour et al., 2017*). Ubiquitination of cytosolic proteins is an established mechanism of substrate-identification by selective autophagy, and TRIF, RIPK1 and RIPK3 are ubiquitinated during inflammatory signaling (*Yang et al., 2017*; *Choi et al., 2018*; *de Almagro et al., 2017*). Although ubiquitinated RIPK1, and RIPK3 were more abundant after LPS/zVAD-induced necroptosis when autophagy was compromised, further investigation is needed to define the components of the selective autophagy machinery that drive the turnover of RHIM-domain proteins, and to determine whether ubiquitination is a critical step in this process.

Necroptosis is acknowledged as a potent pro-inflammatory mode of cell death, but there is an incomplete understanding of its role in normal tissue homeostasis and anti-microbial immunity. Genome-wide association (GWA) and functional studies have revealed that defects in autophagy promote inflammatory diseases such as Crohn's disease, rheumatoid arthritis, lupus and neurodegeneration (*Levine et al., 2011*; *Mizushima, 2007*; *Matsuzawa-Ishimoto et al., 2017*; *Samie et al., 2018*). Our findings provide a potential link between defective autophagy and necroptotic signaling, with autophagy promoting the turnover of RHIM-containing proteins.

# Materials and methods

**Key resources table**

| Reagent type (species) or resource | Designation | Source or reference | Identifiers | Additional information |
|---|---|---|---|---|
| Genetic reagent (*M. musculus*) | Atg16l1loxP/loxP | PMID: 24553140 | | Dr. Aditya Murthy (Genentech, Inc) |
| Genetic reagent (*M. musculus*) | Zbp1loxP/loxP | *Newton et al., 2016* | | Dr. Kim Newton (Genentech, Inc) |
| Commercial assay or kit | Mouse monocyte isolation kit | Miltenyi Biotec | Cat#: 130-100-629 | |
| Peptide, recombinant protein | Cas9 V3 | IDT | Cat#: 1081058 | 10 µg per reaction |
| Peptide, recombinant protein | murine TNFα | Peprotech | Cat#: 315-01A | 50 ng/ml |

*Continued on next page*

Continued

| Reagent type (species) or resource | Designation | Source or reference | Identifiers | Additional information |
|---|---|---|---|---|
| Peptide, recombinant protein | Pam3CSK4 | Invivogen | Cat#: tlrl-pms | 1 µg/ml |
| Peptide, recombinant protein | PolyI:C (LMW) | Invivogen | Cat#: tlrl-picw | 10 µg/ml |
| Peptide, recombinant protein | LPS-EB ultrapure (*E. coli* O111:B4) | Invivogen | Cat#: tlrl-3pelps | 100 ng/ml |
| Peptide, recombinant protein | R848 (Resiquimod) | Invivogen | Cat#: tlrl-r848 | 1 µg/ml |
| Peptide, recombinant protein | CpG-ODN 1826 | Invivogen | Cat#: tlrl-1826 | 5 µM |
| Peptide, recombinant protein | zVAD-fmk | Promega | Cat#: G7232 | 20 µM |
| Chemical compound, drug | Necrostatin-1 | Enzo Life Sciences | Cat#: BML-AP309-0100 | 30 µM |
| Chemical compound, drug | Bafilomycin A1 | Sigma | Cat#: B1793 | 100 nM |
| Chemical compound, drug | MG132 | Sigma | Cat#: M7449 | 2 µM |
| Peptide, recombinant protein | FcR-Block | BD biosciences | Cat#: 5331441 | |
| Chemical compound, drug | Fixable viability dye efluor780 | Invitrogen | Cat#: 65–0865 | |
| Antibody | anti-CD62L PerCP Cy5.5 Rat monoclonal | BD biosciences | Cat#: 560513 RRID: AB_10611578 | Flow cytometry |
| Antibody | anti-CCR2 APC | R and D Systems | Cat#: FAB5538A RRID: AB_10645617 | Flow cytometry |
| Antibody | anti-F4/80 efluor450 Rat monoclonal | eBioscience | Cat#: 48-4801-82 RRID: AB_1548747 | Flow cytometry |
| Antibody | anti-CSF1R BV711 Rat monoclonal | Biolegend | Cat#: 135515 RRID: AB_2562679 | Flow cytometry |
| Antibody | anti-Ly6G BUV395 Rat monoclonal | BD biosciences | Cat#: 565964 RRID: AB_2739417 | Flow cytometry |
| Antibody | anti-CD11b BUV737 Rat monoclonal | BD biosciences | Cat#: 564443 RRID: AB_2738811 | Flow cytometry |
| Antibody | anti-MHCII (IA/IE) PE Rat monoclonal | eBioscience | Cat#: 12-5322-81 RRID: AB_465930 | Flow cytometry |
| Antibody | anti-Ly6C-PECy7 Rat monoclonal | eBioscience | Cat#: 25-5932-82 RRID: AB_2573503 | Flow cytometry |
| Antibody | anti-CD45 FITC Rat monoclonal | eBioscience | Cat#: 11-0451-82 RRID: AB_465050 | Flow cytometry |
| Antibody | anti-F4/80 BV421 Rat monoclonal | Biolegend | Cat#: 123131 RRID: AB_10901171 | Flow cytometry |
| Antibody | anti-CD11b BUV395 Rat monoclonal | BD biosciences | Cat#: 563553 RRID: AB_2738276 | Flow cytometry |
| Antibody | anti-ATG16L1 Mouse monoclonal | MBL international | Cat#: M150-3 RRID: AB_1278758 | Immunoblot |

Continued

| Reagent type (species) or resource | Designation | Source or reference | Identifiers | Additional information |
|---|---|---|---|---|
| Antibody | anti-ATG14L Rabbit polyclonal | MBL international | Cat#: PD026 RRID: AB_1953054 | Immunoblot |
| Antibody | anti-FIP200 Rabbit monoclonal | Cell Signaling Technology | Cat#: 12436 RRID: AB_2797913 | Immunoblot |
| Antibody | anti-Rubicon Mouse monoclonal | MBL international | Cat#: M170-3 RRID: AB_10598340 | Immunoblot |
| Antibody | anti-TRIF Host: Rat | Genentech, Inc | Cat#: 1.3.5 | Immunoblot |
| Antibody | anti-MLKL Host: Rabbit | Genentech, Inc | Cat#: 1G12 | Immunoblot |
| Antibody | anti-p-MLKL Rabbit monoclonal | Abcam | Cat#: ab196436 RRID: AB_2687465 | Immunoblot |
| Antibody | anti-RIPK1 Mouse monoclonal | BD biosciences | Cat#: 610459 RRID: AB_397832 | Immunoblot |
| Antibody | anti-p- RIPK1 Host: Rabbit | Genentech, Inc | Cat#: GNE175.DP.B1 | Immunoblot |
| Antibody | anti-RIPK3 Rabbit polyclonal | Novus Biologicals | Cat#: NBP1-77299 RRID: AB_11040928 | Immunoblot |
| Antibody | anti-p-RIPK3 Host: Rabbit | Genentech, Inc | Cat#: GEN-135-35-9 | Immunoblot |
| Antibody | anti-GSDMD Host: Rat | Genentech, Inc | Cat#: GN20-13 | Immunoblot |
| Antibody | anti-LC3B Rabbit polyclonal | Cell Signaling Technology | Cat#: 2775 RRID: AB_915950 | Immunoblot |
| Antibody | anti-CALCOCO1 Rabbit polyclonal | Proteintech | Cat#: 19843–1-AP RRID: AB_10637265 | Immunoblot |
| Antibody | anti-TAX1BP1 Rabbit monoclonal | Abcam | Cat#: ab176572 | Immunoblot |
| Antibody | anti-p62 Guinea pig polyclonal | Progen biotechnic | Cat#: gp62-c RRID: AB_2687531 | Immunoblot |
| Antibody | anti-NLRP3 Rabbit monoclonal | Cell Signaling Technology | Cat#: 15101 RRID: AB_2722591 | Immunoblot |
| Antibody | anti-ASC Rabbit monoclonal | Cell Signaling Technology | Cat#: 67824 RRID: AB_2799736 | Immunoblot |
| Antibody | anti-STAT1 Rabbit monoclonal | Cell Signaling Technology | Cat#: 14995 RRID: AB_2716280 | Immunoblot |
| Antibody | anti-p- STAT1 Rabbit monoclonal | Cell Signaling Technology | Cat#: 7649 RRID: AB_10950970 | Immunoblot |
| Antibody | anti-M1-polyubiquitin linkage specific antibody | Genentech, Inc | N/A | Immuno precipitation |
| Antibody | anti-K63-polyubiquitin linkage specific antibody | Genentech, Inc | N/A | Immuno precipitation |
| Antibody | Anti-Ubiquitin Mouse monoclonal | Cell Signaling Technology | Cat#: 3936 RRID: AB_331292 | Immunoblot |
| Antibody | anti-beta Actin | Cell Signaling Technology | Cat#: 3700 RRID: AB_2242334 | Immunoblot |
| Antibody | anti-rabbit IgG HRP Goat polyclonal | Cell Signaling Technology | Cat#: 7074 RRID: AB_2099233 | Immunoblot |
| Antibody | anti-mouse IgG HRP Horse polyclonal | Cell Signaling Technology | Cat#: 7076 RRID: AB_330924 | Immunoblot |

*Continued*

| Reagent type (species) or resource | Designation | Source or reference | Identifiers | Additional information |
|---|---|---|---|---|
| Antibody | anti-rat IgG HRP Goat polyclonal | Cell Signaling Technology | Cat#: 7077 RRID: AB_10694715 | Immunoblot |
| Antibody | Anti-Ragweed | Genentech, Inc | N/A | Inhibition |
| Antibody | Anti-mIFNAR1 Mouse monoclonal | Leinco Technologies | Cat#: I-401 RRID: AB_2737538 | Inhibition |
| Antibody | mTNFR2-Fc Mouse Fc | Genentech, Inc | N/A | Inhibition |
| Tools (software) | Image J | | | Immunoblot densitometry |
| Tools (software) | Graphpad Prism 7 | Graphpad | | Data visualization and statistics |

## Mice

Myeloid-specific deletion of *Atg16l1* was achieved by crossing *Lyz2*-Cre + mice with *Atg16l1*$^{loxP/loxP}$ mice (described in 45). Conditional targeting of the *Zbp1* locus was generated in C57BL/6NTac ES cells (Taconic) by introduction of loxP-sites flanking the ATG-containing exon 1, spanning the *Zbp1* 5'UTR and exon one corresponding to NCBI37/mm9 chr2:173,043,537–173,045,687 (described in 12). A 3xFLAG-tag was inserted in-frame with the ATG. Addition of the 3xFLAG-tag did not compromise ZBP1 function, since it failed to rescue the previously described lethality of Ripk1-RHIM mutant mice (*Guo et al., 2018*) (*Supplementary file 1*). Myeloid-specific deletion of *Zbp1* was achieved by crossing *Lyz2*-Cre + mice with *Zbp1*$^{loxP/loxP}$ mice. Combined deletion of *Atg16l1* and *Zbp1* was generated by crossing the above two strains of mice. eGFP-reporter mice were obtained from Jackson labs (strain 57BL/6-Tg(CAG-EGFP)1Osb/J, Stock No: 003291). None of the *in vivo* experiments were randomized. No statistical method was used to pre-determine group sample size, and investigators were not blinded to group allocations or study outcomes.

## LPS-driven sepsis

Intraperitoneal administration of LPS (*E. coli* O111:B4, Sigma L2630) was performed at 10 mg/kg dissolved in a maximum of 200 μL sterile phosphate-buffered saline (PBS). Mice were monitored for morbidity and body temperature every 4 hr for the first 14 hr, followed by monitoring at 24 and 48 hr. Blood was obtained at 4 hr post LPS-administration for serum cytokine analysis. Experiments were performed using age- and sex-matched cohorts from a single colony. All protocols were approved by the Genentech Institutional Animal Care and use Committee; all studies were executed by following relevant ethical regulations detailed in animal use protocols (internal protocol 18–1823).

## Cell culture

Murine monocytes were obtained from femoral bone marrow by negative selection using a monocyte isolation kit (Miltenyi Biotec). Monocyte-derived macrophages were cultured in macrophage medium [high glucose Dulbecco's Minimum Essential Media (DMEM) + 10% FBS+GlutaMAX (Gibco) +Pen/Strep (Gibco) supplemented with 50 ng/ml recombinant murine macrophage-colony stimulating factor (rmM-CSF, Genentech)]. Bone marrow-derived macrophages were generated by culture of total femoral bone marrow in macrophage medium on 15-cm non-TC treated plates for 5 days (Petri dish, VWR). Fresh medium was added on day 3 without removal of original media. On day 5, macrophages were gently scraped from dishes, counted and re-plated on TC treated plates of the desired format for downstream assays. After overnight culture in macrophage medium, assays were performed on day 6 BMDMs. CRISPR-edited BMDMs were treated on day 10 to permit complete protein loss of targeted genes. BMDMs were stimulated with 50 ng/ml murine TNFα (Peprotech), 1 μg/ml Pam3CSK4, 10 μg/ml poly(I:C) LMW, 100 ng/ml ultra-pure LPS unless otherwise stated (LPS, *E. coli* 0111:B4), 1 μg/ml R848 or 5 μM CpG-ODN 1826 (all from Invivogen). zVAD-fmk was added at 20 μM (Promega). Necrostatin-1 (Nec-1) was added at 30 μM (Enzo Life Science). DMSO was added at 0.1% as vehicle control (Sigma). For cell death assays, BMDMs were plated at 2

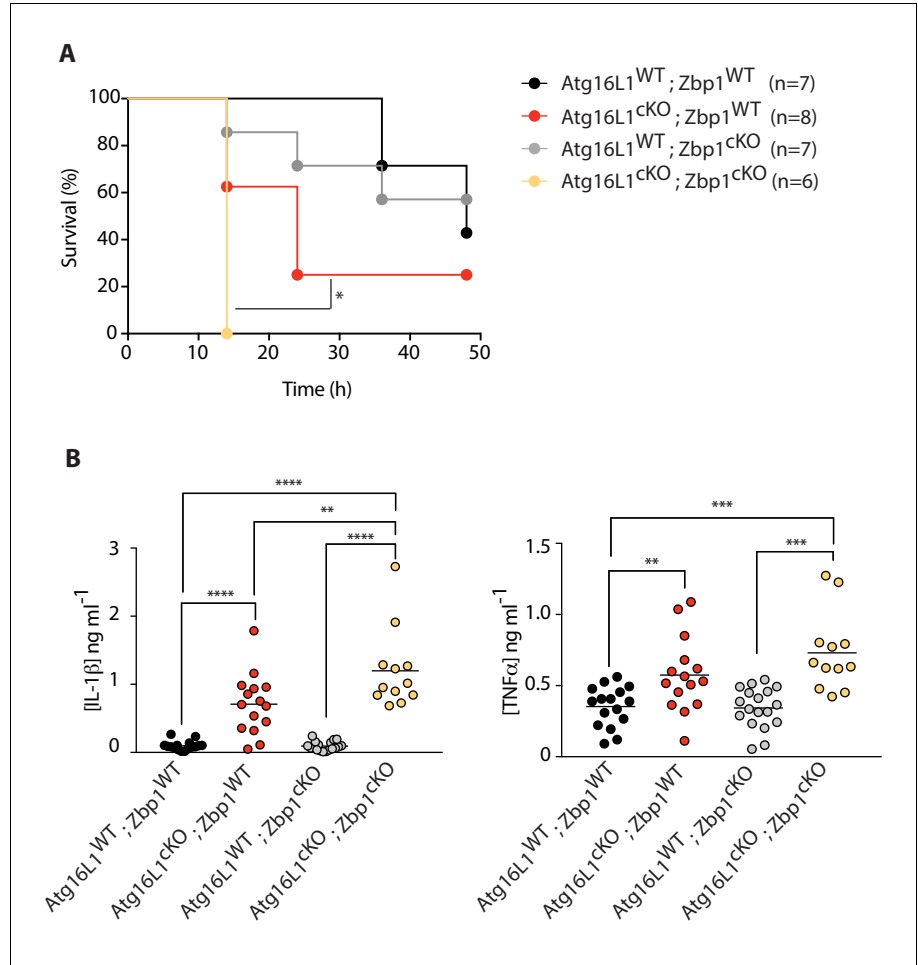

**Figure 7.** Combined loss of myeloid-specific *Atg16l1* and *Zbp1* accelerates LPS-mediated sepsis in mice. (**A**) Kaplan-Meier survival plots for mice following challenge with 10 mg/kg LPS administered intraperitoneally. Statistical analysis *Figure 7—figure supplement 1A* was performed using log-rank test (*Figure 7—figure supplement 1*; *Figure 7—figure supplement 1A*). (**B**) Serum cytokine measurements of IL-1β and TNFα performed by ELISA following 4 hr of intraperitoneal LPS administration at 10 mg/kg. Data in A are representative of two independent experiments. Data in B are pooled from two independent experiments. *p<0.05, **p<0.01, ***p<0.001, ****p<0.0001.

DOI: https://doi.org/10.7554/eLife.44452.027

The following source data and figure supplement are available for figure 7:

**Source data 1.** Accelerated morbidity conferred by double deficiency of ATG16L1 and ZBP1 in myeloid cells following LPS-mediated sepsis in mice.

DOI: https://doi.org/10.7554/eLife.44452.029

**Figure supplement 1.** Accelerated morbidity conferred by double deficiency of ATG16L1 and ZBP1 in myeloid cells following LPS-mediated sepsis in mice.

DOI: https://doi.org/10.7554/eLife.44452.028

$\times$ $10^4$ cells/well in flat-bottom 96-well plates. The following day, cells were stimulated as indicated to induce necroptosis. BMDM viability was assessed by propidium iodide (PI, Invitrogen) staining using live-cell imaging, measuring PI-positive cells per $mm^2$ (Incucyte systems, Essen Biosciences). Percent cell death was calculated by dividing PI-positive cells per $mm^2$ with total plated cells per $mm^2$. Total cell plated cells were enumerated by independently plating BMDMs and staining with a nuclear dye fluorescing in the same channel as PI (Nuclear-ID Red, Enzo Life Science) or addition of 0.1% Triton X-100 in the presence of PI. Time points between 12 and 16 hr were used to compare cell death unless otherwise stated. ZBP1 turnover measured by cycloheximide-chase assays was

performed with 100 µg/ml CHX in DMSO for the indicated time points (Sigma). ZBP1 degradation assays in WT BMDMs were performed using 100 nM Bafilomycin A1 (Sigma) or 2 µM MG132 (Sigma) for the indicated time points.

## Gene editing
CRISPR/Cas9-mediated deletion of genes was performed by electroporation of Cas9 RNP in monocytes and BMDMs. Briefly, $5 \times 10^6$ primary monocytes from the bone marrow or day five cultured BMDMs were electroporated with recombinant Cas9 (IDT) complexed with gene-specific guide RNAs (*Supplementary file 2*). Briefly, locus-specific crRNAs were annealed with tracrRNAs at a 1:1 stoichiometric ratio at 95°C for 5 min followed by complexing with recombinant Cas9 at a 3 µL:1 µL gRNA:Cas9 ratio per guide RNA to generate the RNP complex. two guide RNAs were combined per gene. Cells were resuspended in 20 µL nucleofector solution P3 and RNP complex added. This mixture was aliquoted into 16-well nucleofector strips (Lonza) and electroporated using program CM-137 (4D-Nucleofector, Lonza). Following electroporation, cells were grown in non-tissue culture treated dishes (VWR) for an additional 5 days in macrophage media. On day 5, cells were scraped from dishes and re-plated as required for functional assays in tissue-culture treated multi-well plates.

## Flow cytometry
Monocyte-derived macrophages or BMDMs were harvested and washed in cold PBS. Cells were incubated in Fc-block reagent (BD biosciences) and fixable viability dye eFluor 780 (Invitrogen) for 15 min at 4°C in cold PBS. Monocyte-derived macrophages were washed once and stained with the following antibodies: anti-CD62L PerCp-Cy5.5 (BD biosciences), anti-CCR2 APC (R and D Systems), anti-F4/80 efluor450 (eBioscience), anti-CSF1R BV711 (Biolegend), anti-Ly6G BUV395 (BD biosciences), anti-CD11b BUV737 (BD biosciences), anti-MHCII(I-A/I-E) PE (eBioscience) and anti-Ly6C PE-Cy7 (eBioscience). BMDMs were washed once and stained with the following antibodies: anti-CD45 FITC (eBioscience), anti-F4/80 BV421 (Biolegend), anti-CD11b BUV395 (BD biosciences). Stained cells were analyzed on by flow cytometry using a BD FACS CANTO instrument. Loss of eGFP or CD45 was assessed by gating on live F4/80 + macrophages using FlowJo X.

## ELISA
BMDM cell culture medium or murine serum was analyzed for measurement of cytokines IL-1β and TNFα (eBioscience) by ELISA following manufacturer's protocols.

## Immunoblotting
To assay CRISPR-mediated gene deletions, cell pellets were lysed in RIPA buffer +protease and phosphatase inhibitors (Roche). Supernatants were obtained after high speed centrifugation and protein concentration measured using the BCA assay (Thermo Fisher). To perform detergent soluble and insoluble fractionation, cell pellets were lysed in 1% NP-40 lysis buffer (150 mM NaCl, 20 mM Tris-HCl pH 7.5, 1% NP-40, 1 mM EDTA, protease and phosphatase inhibitors). Lysates were flash frozen on dry-ice, thawed on ice and vortexed for 10 s followed by centrifugation at 1000 g for 10 min to remove nuclear pellets. Supernatants were centrifuged at 15000 rpm (or highest speed) for 15 min in a refrigerated table-top centrifuge. Resultant supernatants were collected as NP-40 soluble fractions. NP-40 insoluble pellets were resuspended in 1% NP-40 lysis buffer supplemented with 1% SDS. The suspension was homogenized by passing through a 26-gauge needle, and protein quantification performed using BCA assay. To assess ubiquitination, cells were lysed under denaturing conditions with lysis buffer (9 M urea and 20 mM HEPES, pH 8.0) containing 1 mM sodium orthovanadate, 2.5 mM sodium pyrophosphate, 1 mM beta-glycerolphosphate and incubated for 20 min with vigorous shaking at 900 rpm at room temperature. Following incubation, cell lysates were centrifuged for 10 min at 14,000 rpm. Lysates were then diluted two times with buffer (20 mM HEPES, pH 8.0) containing Roche protease inhibitor cocktail, 100 µM PR-619 (SI9619, Life Sensors) and 100 µM 1,10-phenanthroline (SI9649, Life Sensors) and used for immunoprecipitation with ubiquitin chain-specific antibodies and protein-A/G beads overnight at 4°C as previously described in *Goncharov et al. (2018)*. SDS-PAGE was performed using a 4–12% gradient Bis-Tris gel (Novex), followed by protein transfer onto PVDF membranes and antibody incubation. Immunoblots were detected by enhanced chemiluminescence (western lightning-plus ECL, Perkin Elmer). Antibodies

used: anti-Atg16l1 (MBL international), anti-Atg14l (MBL international), anti-FIP200 (Cell Signaling Technology), anti-Rubicon (MBL), anti-TRIF (Genentech), anti-MLKL (Genentech), anti-RIPK1 (BD Biosciences), anti-pSer345 MLKL (Abcam), anti-pSer166/Thr169 RIPK1 (Genentech), anti-RIPK3 (Novus Biologicals), anti-pThr231/Ser232 RIPK3 (Genentech), anti-GSDMD (Genentech,), anti-LC3B (Cell Signaling Technology), anti-Calcoco1 (Proteintech), anti-Tax1bp1 (Abcam), Sqstm1/p62 (Progen biotechnic), anti-NLRP3 (Cell Signaling Technology), anti-ASC (Cell Signaling Technology), anti-STAT1 (Cell Signaling Technology), anti-pTyr701 STAT1 (Cell Signaling Technology), anti-M1 polyubiquitin linkage-specific antibody (Genentech), anti-K63 polyubiquitin linkage-specific antibody (Genentech), anti-β-actin (Cell Signaling Technology), anti-rabbit IgG-HRP (Cell Signaling Technology), anti-mouse IgG-HRP (Cell Signaling Technology), anti-rat-HRP Ig (Cell Signaling Technology). ImageJ was used to quantify immunoblot density.

## Statistical analysis

Pairwise statistical analyses were performed using an unpaired Student's two-sided t-test to determine if the values in two sets of data differ. Correction for multiple-comparisons was performed using the Holm-Sidak method with $\alpha$ = 0.05. Scatterplot bars and connected dot plots present means of data. Analysis of kinetic (time) or LPS dose-response datasets were performed using two-way ANOVA followed by multiple comparison testing. Line graphs and associated data points represent means of data; error bars represent standard deviation from mean. For LPS-mediated sepsis studies, a log-rank (Mantel-Cox) test was used to assess significance of the differences between indicated groups in their survival. GraphPad Prism seven was used for data analysis and representation.

# Acknowledgements

We thank K Cherry, T Scholl, B Torres and W Ortiz for animal husbandry, K Wickliffe, K Rajasekaran, K Heger and E Freund for technical assistance, and T Yi, M Albert, S Turley and I Mellman for critical review of this work.

# Additional information

#### Competing interests

Junghyun Lim, Hyunjoo Park, Jason Heisler, Timurs Maculins, Merone Roose-Girma, Min Xu, Brent Mckenzie, Menno van Lookeren Campagne, Kim Newton, Aditya Murthy: Affiliated with Genentech. No other competing interests to declare.

#### Funding

| Funder | Author |
|--------|--------|
| Genentech | Aditya Murthy |

The funders had no role in study design, data collection and interpretation, or the decision to submit the work for publication.

#### Author contributions

Junghyun Lim, Conceptualization, Data curation, Formal analysis, Validation, Investigation, Methodology, Writing—review and editing; Hyunjoo Park, Jason Heisler, Investigation; Timurs Maculins, Validation, Investigation, Writing—review and editing; Merone Roose-Girma, Resources; Min Xu, Resources, Supervision, Methodology, Project administration; Brent Mckenzie, Resources, Supervision, Project administration; Menno van Lookeren Campagne, Supervision, Writing—original draft, Writing—review and editing; Kim Newton, Resources, Supervision, Methodology, Writing—original draft, Project administration, Writing—review and editing; Aditya Murthy, Conceptualization, Data curation, Formal analysis, Supervision, Investigation, Visualization, Methodology, Writing—original draft, Project administration, Writing—review and editing

## Author ORCIDs

Aditya Murthy  https://orcid.org/0000-0002-6130-9568

## Ethics

Animal experimentation: All protocols were approved by the Genentech Institutional Animal Care and use Committee; all studies were executed by following relevant ethical regulations detailed in animal use protocols.(internal protocol 18-1823).

## Decision letter and Author response

Decision letter https://doi.org/10.7554/eLife.44452.035
Author response https://doi.org/10.7554/eLife.44452.036

## Additional files

### Supplementary files

• Supplementary file 1. Perinatal lethality of *Ripk1*^*RHIM/RHIM*^ mice is prevented by ZBP1 deficiency (*Newton et al., 2016*) but not by addition of a 3xFlag N-terminal tag to ZBP1.
DOI: https://doi.org/10.7554/eLife.44452.030

• Supplementary file 2. crRNA targeting sequences used for CRISPR/Cas9 gene editing.
DOI: https://doi.org/10.7554/eLife.44452.031

• Transparent reporting form
DOI: https://doi.org/10.7554/eLife.44452.032

### Data availability

All data generated or analyzed during this study are included in the manuscript and supporting files.

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
