## [Decision Letter]

Thank you for submitting your article "Autophagy regulates inflammatory programmed cell death via turnover of RHIM-domain proteins" for consideration by *eLife*. Your article has been reviewed by three peer reviewers, and the evaluation has been overseen by a Reviewing Editor and Tadatsugu Taniguchi as the Senior Editor. The following individuals involved in review of your submission have agreed to reveal their identity: Rupert Beale (Reviewer #2); Katherine Fitzgerald (Reviewer #3).

The reviewers have discussed the reviews with one another and the Reviewing Editor has drafted this decision to help you prepare a revised submission. As you will see from their reports, all three reviewers have found the manuscript interesting and worthy of inviting for a re-submission. However, they have raised several points that need to be addressed to increase the general impact of the work. In particular, there was a general concern among reviewers to better address the mechanism.

1) The authors should perform a time course of the process comparing WT vs. *Atg16L1* KO to substantiate their claim that autophagy is important to attenuate/resolve necroptosis in their system. In this time course, different parameters must be analyzed –% cell death, accumulation of detergent-insoluble forms of TRIF, RIPK1 and RIPK3 among others.

2) Autophagy is presented as an essential process, however the mechanism is only studied according to the consequences of knocking down different autophagy proteins. The authors should characterize autophagy in more detail by monitor autophagy flux after cell death induction (LC3 lipidation, p62 levels, mTORC1 activation among others).

3) The authors should provide more evidence as to the type of death – necroptosis vs. pyroptosis.

4) Figure 3 shows accumulation of detergent-insoluble forms of TRIF, RIPK1, RIPK3, We consider that detecting ubiquitination of these proteins will support their hypothesis. An analysis of the role of TaxBP1 (and other autophagy receptors) would greatly strengthen the manuscript.

5) The comments on the statistics need to be addressed.

I am adding the reviewers' comments so that you get an idea of their different views as well as experimental work that you might carry out to add to accommodate their concerns.

*Reviewer #1:*

In this paper, Lim et al., proposes a novel mechanism in macrophages, in which autophagy, promoting the turnover of RHIM-domain proteins (RIPK1, RIPK3, TRIF and ZBP1), induces the suppression/attenuation of necroptosis, a type of necrotic cell death.

Necrotic cell death (necroptosis and Pyroptosis) protect host against microbial pathogens and dysregulation of this process has been involved in autoimmune and auto-inflammatory conditions. In our opinion authors try to address an important question in this context: what is the mechanism that drives returning to homeostasis after necrotic cell death induction?. They do so mainly doing extensive in vitro work, claiming that autophagy attenuates necroptosis through the selective degradation of RHIM-domain proteins. To reach this conclusion they use a very well-known model: conditional *Atg16L* KO mice lacking this key autophagy protein in myeloid cells.

However, although the work is interesting and with high potential to end up being an important contribution to the field, we find that authors overestimated their results making too simple conclusions. In general:

- They claim that the type of necrotic cell death that is being detected/quantified is Necroptosis. In our opinion a deeper analysis is required to make this claim. Considering the crosstalk between necroptosis and pyroptosis, we think that authors have oversimplified their conclusions.

- They make a strong statement describing canonical autophagy as a mechanism driving the attenuation of necroptosis. Although it is potentially interesting, especially considering the current interest in selective autophagy, authors must improve this part of the work.

Therefore, authors have to revise the work obtaining convincing data to support their main claims. We think that the following points have to be addressed to make this work suitable for publication:

- Authors quantify% of cell death, and they conclude that this is necroptosis. We do understand that using the combination of TNF/LPS+zVAD is a well-established system to induce this type of death cell in the context of WT cells. However, it is not known if this increase of% death cells in *Atg16L* KO cells is necroptosis or pyroptosis. Considering the crosstalk among these two processes, the existence of Gsdmd-independent pyroptosis (Schneider et al., 2017), and the failure rescue with Nec-1 in the case of LPS, we think that the authors should further characterize this kind of cell death.

- Authors claim that autophagy is important to attenuate/resolve necroptosis in their system. To do this claim, we think that it is necessary to do a time course of the process comparing WT vs. *Atg16L1* KO. In this time course, different parameters must be analyzed –% cell death, accumulation of detergent-insoluble forms of TRIF, RIPK1 and RIPK3 among others.

- Autophagy is presented as an essential process. This mechanism is only studied according to the consequences of knocking down different autophagy proteins. However, we consider that authors should describe autophagy in more detail. They could monitor autophagy flux after cell death induction. Actually, an important issue in all this work is the fact of obtaining different results with TLR3 and TLR4. Maybe autophagy flux is different in these two contexts.

- Authors see an accumulation of RHIM-domain proteins in *Atg16L* KO cells. They should try to recapitulate this accumulation using lysosome inhibitors to prove that autophagy core proteins are being used in a lysosomal degradation process (they do so for ZBP1, Figure 6—figure supplement 1).

- As we highlighted before, the technique used to induce necroptosis is well established in the field. However, it is known that this method induces autophagy. Would it be possible to use another technique to induce necroptosis, without targeting autophagy?

- Figure 3 shows accumulation of detergent-insoluble forms of TRIF, RIPK1 and RIPK3. Authors comment in the Discussion (second paragraph) about the possibility of protein ubiquitination playing a role in this process. We consider that detecting ubiquitination of these proteins will support their hypothesis considering that authors make a strong statement about the role of selective autophagy in this process.

*Reviewer #2:*

This manuscript extends on observations reported by this laboratory in Samie et al., 2018, which reported Tax1BP1 mediated selective autophagy of TRIF. Here, the authors demonstrate that oligomerised RHIM domain proteins (including TRIF) accumulate in ATG16L1 deficient macrophages. They investigate autophagy as a possible homeostatic mechanism for clearing these oligomers, thereby opposing inflammation and cell death. Consistent with this, loss of essential autophagy genes resulted in enhanced cell death in BMDMs stimulated with LPS or Poly I:C and zVAD. A notable strength of this paper which distinguishes it from previous observations on RIP kinases and autophagy is that the authors provide evidence of involvement of the FIP200 and ATG14, which indicates this is due to canonical macroautophagy and not a 'non-canonical' or LAP-like process. The loss of Rubicon makes no difference, and whilst this does not rule out a contribution from 'non-canonical' autophagy (see discussion in Fletcher, Ulferts et al. EMBOJ 2018), taken together with the other genetic data it strongly argues against LAP being required here. The data to support this are shown in Figures 1F and Supplementary Figure 2B, the paper would be much easier to read if these data were grouped together.

The authors demonstrate that Necrostatin fails to completely suppress necroptosis in autophagy deficient cells, and show this depends on Mlkl and Ripk3. ATG16L1 deficient cells accumulate detergent insoluble forms of RHIM domain proteins, consistent with a role for autophagy in their degradation. The authors go on to show that ZBP1 accumulates in ATG16L1 deficient cells, but that curiously this protects against necroptosis. This is interesting, but the effect only seems to take place in the setting of ATG16L1 deficiency so it remains unclear if there is a physiological relevance.

Absent from this paper are any direct measurements of autophagy or autophagic flux. It would be very helpful to the authors' argument to demonstrate that oligomerised RHIM-domain proteins do indeed become selectively engulfed by autophagosomes. As a minimum I would expect to see data on accumulation of autophagosomes and associated lipidation of LC3 after RHIM-domain oligomerisation, and (crucially) the effect of ATG14/FIP200/ATG16L1 deletion on this process. This would provide additional strong evidence of regulation of signalling by canonical autophagy. I accept the authors argument that a full mechanistic explanation of exactly how the proteins are ubiquitinated and degraded (assuming this is the case) is beyond the scope of the paper, but given their previous findings of Tax1BP1 being a critical component required for degradation of TRIF (at least in mice, the situation in humans may well be different), it is surprising that the authors do not comment at all on this. It would greatly strengthen the paper to see data on the requirement (or otherwise) of autophagy receptors such as TaxBP1, p62 and (in humans) NDP52.

In summary this paper provides an important if somewhat limited advance in our understanding of the regulation of necroptosis. The conclusions are in general well supported by the data. The paper would be greatly strengthened by exploring the mechanisms of selective autophagy in this setting.

Additional data files and statistical comments:

Many of the pairwise analyses throughout require correction for multiple hypotheses.

*Reviewer #3:*

In this manuscript, Lim et al. show that autophagy is involved in clearing activation of the necroptotic machinery in order to limit cell death. Inhibition of autophagy leads to increased cell death following ligand /caspase inhibition. This was shown to be caused by increased oligomerization of necroptotic components and reduced clearance. Interestingly, whereas ZBP1 has previously been shown to be capable of triggering necroptosis, loss of ZBP1 in autophagy-deficient cells lead to further increase in death, suggesting that in this scenario, ZBP1 keeps necroptosis in check. The authors further show that mice lacking canonical autophagy and ZBP1 do worse in a model of septic shock.

The paper makes a valuable contribution to understanding the regulatory machinery of cell death pathways and reveals the importance of autophagy in limiting necroptosis. They also show that when autophagy is perturbed, necroptosis can progress independently of RIPK1. Overall the paper is well written and the data very compelling. However some additional insights on how ZBP1 can repress necroptosis would strengthen this study.

– The authors should provide some insights into how ZBP1 functions in repressing necroptosis.

– Ripk1 inhibition could not completely block necroptosis when TRIF is involved (TLR3 and TLR4 stimuli in autophagy deficient cells). Can RIPK3 be recruited to the TRIFosome and activated in the absence of RIPK1? Is it just the kinase activity of RIPK1 that becomes redundant (i.e. is scaffolding still required)? Use of RIPK1 sgRNA could address this.

– How is necroptosis triggered in non-TRIF mediated TLR triggered necroptosis? Through autocrine secretion of TNF or IFN?

– The recent paper (Sarhan et al., 2018) suggests that autocrine IFN is required for necroptosis. Is IFN involved in necroptosis when autophagy is not functional? Does perturbed clearance of TRIF and RIPK1 result in higher IFN signaling and thus higher cell death?

– Figure 3D and F suggests that when autophagy is deranged, RIPK1- and RIPK3- oligomerization is dependent on TRIF but not phosphorylation of RIPK1 and RIPK3?

– The term "screening" is a bit misleading in Figure 1. They optimized electroporation conditions for delivery of sgRNA:Cas9 RNPs into primary BMDMs. This could be added to the supplementary figure.

---

## [Author Response]

As you will see from their reports, all three reviewers have found the manuscript interesting and worthy of inviting for a re-submission. However, they have raised several points that need to be addressed to increase the general impact of the work. In particular, there was a general concern among reviewers to better address the mechanism.1) The authors should perform a time course of the process comparing WT vs. Atg16L1 KO to substantiate their claim that autophagy is important to attenuate/resolve necroptosis in their system. In this time course, different parameters must be analyzed –% cell death, accumulation of detergent-insoluble forms of TRIF, RIPK1 and RIPK3 among others.

We have performed time-course analysis of necroptosis in WT vs. *Atg16l1*-cKO BMDMs by LPS or PolyI:C. These demonstrate accelerated necroptosis in *Atg16l1*-cKO BMDMs over 18 hours of treatment (Figure 4, Figure 4—figure supplement 1B, C). Interestingly, PolyI:C alone exhibits elevated death of *Atg16l1*-cKO BMDMs (Figure 4—figure supplement 1C, PolyI:C time-course). This demonstrates that TLR3, which only signals through TRIF, can induce death of autophagy-deficient macrophages and is consistent with the observed over-accumulation of TRIF in these cells. We also show accumulation of detergent-insoluble forms of TRIF (Figure 4B), autophosphorylated and total RIPK1 (Figure 4C), and autophosphorylated and total RIPK3 (Figure 4D) in both wild-type and *Atg16l1*-cKO BMDMs over 6 hours of LPS/zVAD treatment. Loss of *Atg16l1* enhances accumulation of active TRIF, RIPK1 and RIPK3 with varying kinetics. These new findings are discussed in the second paragraph of the subsection “Loss of autophagy results in accumulation of active forms of TRIF, RIPK1 and RIPK3 during necroptosis”.

2) Autophagy is presented as an essential process, however the mechanism is only studied according to the consequences of knocking down different autophagy proteins. The authors should characterize autophagy in more detail by monitor autophagy flux after cell death induction (LC3 lipidation, p62 levels, mTORC1 activation among others).

We analyzed wild-type BMDMs in a time-course over 6 hours using the well-established method of halting lysosomal degradation by the lysosomal inhibitor Bafilomycin A1 to block autophagic flux. It is not possible to study kinetics of autophagy in ATG16L1 deficient cells since they cannot perform autophagy. We observe a rapid accumulation of lipidated LC3B (LC3-II), a marker of autophagosome maturation, during LPS/zVAD mediated necroptosis in the presence of Bafilomycin A1 (Figure 4—figure supplement 2G). Importantly, selective autophagy receptors SQSTM1/p62, CALCOCO1 and TAX1BP1 preferentially accumulate in detergent-insoluble fractions when lysosomal function is blocked (compare Figure 4—figure supplement 2G vs. 2H). These new findings are discussed in the subsection “The autophagy receptor TAX1BP1 prevents TRIF-mediated necroptosis”.

Along with the above autophagy receptors, we now show that active TRIF, RIPK1 and RIPK3 also accumulate in detergent-insoluble fractions of wild-type BMDM lysates when lysosomal function is inhibited (Figure 4—figure supplement 2A-C). In contrast, basal turnover of TRIF, RIPK1 and RIPK3 does not seem to depend on lysosomal turnover via autophagy (Figure 4—figure supplement 2D-F). Together, these data clearly demonstrate that autophagic flux and lysosomal function are critical for the turnover of active forms of TRIF, RIPK1 and RIPK3 during necroptosis. These are discussed in the last paragraph of the subsection “Loss of autophagy results in accumulation of active forms of TRIF, RIPK1 and RIPK3 during necroptosis”.

None of our studies were performed under nutrient-deprived conditions used to induce mTOR-mediated autophagy. We thus focused our analysis on markers of autophagosome maturation and selective autophagy. The investigation of starvation-induced signals (i.e. mTOR, AKT) are outside the scope of the current manuscript. We believe that our efforts provide sufficient evidence to demonstrate that autophagic flux is induced in wild-type BMDMs during necroptosis.

3) The authors should provide more evidence as to the type of death - necroptosis vs. pyroptosis.

Reviewer #1 proposed that we investigate secondary pyroptosis as it was recently demonstrated to drive Gsdmd-independent pyroptosis. Schneider et al., 2017, identified NLRP3 and ASC were critical drivers of GSDMD-independent pyroptosis in BMDCs. Thus, in addition to our previous demonstration that RIPK1-independent BMDM death depended on RIPK3 and MLKL, we have now generated *Nlrp3* and *Pycard/ASC* knockdown by CRISPR/Cas9 (Figure 2—figure supplement 1B). Loss of *Nlrp3* or *Pycard/ASC* did not protect BMDMs from death by LPS/zVAD or PolyI:C/zVAD. Furthermore, it did not fully rescue RIPK1-independent death (Figure 2—figure supplement 1C). These data are consistent with our previous observations that necroptosis driven by RIPK3 and MLKL is the relevant type of cell death in our experimental system. We discuss these new findings in the first paragraph of the subsection “TRIF and RIPK1 drive necroptosis in *Atg16l1*-deficient macrophages”.

4) Figure 3 shows accumulation of detergent-insoluble forms of TRIF, RIPK1, RIPK3, We consider that detecting ubiquitination of these proteins will support their hypothesis. An analysis of the role of TaxBP1 (and other autophagy receptors) would greatly strengthen the manuscript.

Detecting ubiquitination of detergent-insoluble proteins is a technically challenging assay due to the harsh denaturing conditions (9M Urea) required. Nonetheless, we have successfully measured ubiquitination of RIPK1 and RIPK3 in primary BMDMs under these conditions. Using a mixture of M1 and K63-linkage-specific antibodies, we successfully immunoprecipitated autophosphorylated RIPK1 and RIPK3 from wild-type and *Atg16l1*-cKO BMDMs following LPS/zVAD treatment. As shown in Figure 4E, we demonstrate that loss of *Atg16l1* results in accumulation of poly-ubiquitinated forms of autophosphorylated RIPK1 and RIPK3 upon LPS/zVAD treatment. This is consistent with our previous observations that high molecular weight forms of autophosphorylated RIPK1 and RIPK3 accumulate in *Atg16l1*-cKO BMDMs (Figure 3D, F; Figure 4C, D). These data are discussed in the second paragraph of the subsection “Loss of autophagy results in accumulation of active forms of TRIF, RIPK1 and RIPK3 during necroptosis”.

In line with findings in response to question 2, we demonstrate that loss of *Atg1l1* phenocopies Bafilomycin A1-mediated lysosomal inhibition, since SQSTM1/p62, TAX1BP1 and CALCOCO1 accumulate in detergent-insoluble fractions of *Atg16l1*-cKO BMDMs to a greater degree than wild-type controls over 6 hours of necroptosis by LPS/zVAD (Figure 5B). To demonstrate a functional role for autophagy receptors in BMDM necroptosis, we have successfully generated CRISPR-mediated knockdown of *Sqstm1/p62, Tax1bp1* or *Calcoco1* in wild-type BMDMs (Figure 5C). These autophagy receptors were previously identified as candidate receptors accumulating in *Atg16l1*-cKO BMDMs (Samie et al., 2018). Induction of LPS/zVAD or PolyI:C/zVAD-mediated necroptosis revealed that knockdown of *Tax1bp1* significantly increased BMDM necroptosis (Figure 5D). Consistent with the phenotypes of BMDMs lacking ATG16L1, ATG14L or FIP200, TAX1BP1 deficiency resulted in Necrostatin-1 resistant cell death following activation of TRIF signaling via TLR3 or TLR4 (Figure 5D, PolyI:C+zVAD+Nec-1 or LPS+zVAD+Nec-1 conditions). Together, these new data demonstrate that autophagic flux is rapidly induced as a mechanism of RHIM-protein clearance during necroptosis. The selective autophagy receptor TAX1BP1 appears important in this pathway, since its deletion in wild-type BMDMs phenocopies autophagy deficiency. These data are described in the subsection “The autophagy receptor TAX1BP1 prevents TRIF-mediated necroptosis”.

5) The comments on the statistics need to be addressed.

We have revised our pairwise analyses to correct for multiple comparisons (Holms-Sidack method, α=0.05). This has not impacted any conclusions in the original submission. We have amended our Materials and methods appropriately to reflect this revision (subsection “Statistical analysis”).

I am adding the reviewers' comments so that you get an idea of their different views as well as experimental work that you might carry out to add to accommodate their concerns.Reviewer #1:[…] Although the work is interesting and with high potential to end up being an important contribution to the field, we find that authors overestimated their results making too simple conclusions. In general:- They claim that the type of necrotic cell death that is being detected/quantified is Necroptosis. In our opinion a deeper analysis is required to make this claim. Considering the crosstalk between necroptosis and pyroptosis, we think that authors have oversimplified their conclusions.- They make a strong statement describing canonical autophagy as a mechanism driving the attenuation of necroptosis. Although it is potentially interesting, especially considering the current interest in selective autophagy, authors must improve this part of the work.Therefore, authors have to do a better work obtaining convincing data to support their main claims. We think that the following points have to be addressed to make this work suitable for publication:- Authors quantify% of cell death, and they conclude that this is necroptosis. We do understand that using the combination of TNF/LPS +z VAD is a well-established system to induce this type of death cell in the context of WT cells. However, it is not known if this increase of% death cells in Atg16L KO cells is necroptosis or pyroptosis. Considering the crosstalk among these two processes, the existence of Gsdmd-independent pyroptosis (Schneider et al., 2017), and the failure to rescue with Nec-1 in the case of LPS, we think that the authors should further characterize this kind of cell death.

We thank the Reviewer for this comment. RIPK3 and MLKL are well-established as bona-fide gatekeepers of necroptosis, and in our initial submission we demonstrated that these genes were required for death of *Atg16l1*-cKO BMDMs, even under RIPK1-independent (Nec-1 resistant) conditions. Nonetheless, we acknowledge that additional modes of death could account for the Nec-1 resistant phenotype. In our revised manuscript, we provide additional evidence to support our conclusion that the type of cell death observed in our study is necroptosis. CRISPR-mediated knockdown of *Nlrp3* or *Pycard/ASC,* recently demonstrated to drive GSDMD-independent pyroptosis (Schneider et al., 2017), did not rescue Nec-1 resistant death of *Atg16L1*-cKO BMDMs (Figure 2—figure supplement 1B, C). In contrast, knockdown of *Ripk3* or *Mlkl* fully rescued Nec-1 resistant death of *Atg16l1*-cKO BMDMs (Figure 2B, Figure 2—figure supplement 1A). Together with the lack of rescue observed by *Gsdmd* deletion (Figure 2B, Figure 2—figure supplement 1A), we believe we have stronger evidence to support necroptosis as the primary form of cell death in our study. These data are discussed in the first paragraph of the subsection “TRIF and RIPK1 drive necroptosis in *Atg16l1*-deficient macrophages”.

- Authors claim that autophagy is important to attenuate/resolve necroptosis in their system. To do this claim, we think that it is necessary to do a time course of the process comparing WT vs. Atg16L1 KO. In this time course, different parameters must be analyzed –% cell death, accumulation of detergent-insoluble forms of TRIF, RIPK1 and RIPK3 among others.

The studies prompted by this comment proved highly informative in better describing the process of necroptosis in *Atg16l1*-cKO BMDMs. Thus, we have generated a new main figure (Figure 4) and supplementary figure (Figure 4— figure supplement 1). In these new datasets, we describe accelerated death of *Atg16l1*-cKO BMDMs following necroptosis induced by LPS or PolyI:C. Loss of *Atg16l1* resulted in >80% cell death by 3 hours in both settings, compared to approximately 50% of wild-type BMDMs (Figure 4A; Figure 4—figure supplement 1B, C). We thus focused our time course to 6 hours of treatment, and show that TRIF, RIPK1 and RIPK3 accumulate in detergent-insoluble fractions with distinct kinetics. TRIF accumulates rapidly and transiently, with a greater accumulation observed in *Atg16l1*-cKO BMDMs (Figure 4B). Autophosphorylated and total RIPK1 accumulate at 2 hours, with sustained and increased levels in *Atg16l1*-cKO BMDMs (Figure 4C). Autophosphorylated and total RIPK3 accumulates with different kinetics between wild-type and *Atg16l1*-cKO BMDMs. Whereas maximal levels are observed at 6 hours in wild-type BMDMs, loss of *Atg16l1* accelerates this process. Maximum levels of p-RIPK3 and total RIPK3 are observed at 2 hours of LPS/zVAD treatment (Figure 4D). These new datasets provide insight into the accelerated cell death of BMDMs during defective autophagy.

- Autophagy is presented as an essential process. This mechanism is only studied according to the consequences of knocking down different autophagy proteins. However, we consider that authors should describe autophagy in more detail. They could monitor autophagy flux after cell death induction. Actually, an important issue in all this work is the fact of obtaining different results with TLR3 and TLR4. Maybe autophagy flux is different in these two contexts.

We provide new datasets describing autophagic flux during necroptosis in wild-type BMDMs (loss of *Atg16L1* prevents analysis of autophagic flux in these cells). Lysosomal inhibition with Bafilomycin A1 is a well-established method to perturb autophagic flux and permit the accumulation of autophagic cargo destined for lysosomal degradation. By adding Bafilomycin A1 during LPS/zVAD mediated necroptosis, we observe rapid accumulation of lipidated LC3B (LC3-II), a hallmark of autophagosome biogenesis (Figure 4—figure supplement 2G). Additionally, autophagy receptors are known to traffic to the lysosome to regulate turnover of autophagic cargo. Consistent with this, we observe accumulation of specific autophagy receptors SQSTM1/p62, TAX1BP1 and CALCOCO1 in detergent-insoluble fractions of necroptotic wild-type BMDMs in the presence of Bafilomycin A1 (Figure 4—figure supplement 2G). Loss of *Atg16l1* phenocopies Bafilomycin A1-mediated lysosomal inhibition, since SQSTM1/p62, TAX1BP1 and CALCOCO1 accumulate in detergent-insoluble fractions of *Atg16l1*-cKO BMDMs to a greater degree than wild-type controls over 6 hours of necroptosis` by LPS/zVAD (Figure 5B). Together, these findings more conclusively demonstrate that autophagy is rapidly induced in BMDMs during necroptosis. We discuss these findings in the subsection “The autophagy receptor TAX1BP1 prevents TRIF-mediated necroptosis”.

Comparing kinetics of cell death by TLR3 and TLR4 stimulation revealed that PolyI:C was sufficient to reveal enhanced cell death in *Atg16L1*-cKO BMDMs even in the absence of zVAD-fmk (Figure 4—figure supplement 1B, LPS only vs. Figure 4—figure supplement 1C, PolyI:C only). This is consistent with the knowledge that TLR3 only signals through TRIF, resulting in the generation of a death-inducing complex termed the RIPoptosome (Kaiser, 2013). While TLR4 is capable of this as well, it can engage MYD88 to promote pro-survival pathways that antagonize RIPoptosome activity. We thank the reviewer for this comment and discuss these points in the second paragraph of the subsection “Loss of autophagy results in accumulation of active forms of TRIF, RIPK1 and RIPK3 during necroptosis” and in the first paragraph of the subsection “TRIF and RIPK1 drive necroptosis in *Atg16l1*-deficient macrophages”.

- Authors see an accumulation of RHIM-domain proteins in Atg16L KO cells. They should try to recapitulate this accumulation using lysosome inhibitors to prove that autophagy core proteins are being used in a lysosomal degradation process (they do so for ZBP1, Figure 6—figure supplement 1).

In order to demonstrate that lysosomal degradation is responsible for the turnover of TRIF, RIPK1 and RIPK3, we induced LPZ/zVAD-mediated necroptosis in wild-type BMDMs in the presence of Bafilomycin A1. Over a 6-hour time-course, we observed accumulation of high molecular weight forms of TRIF, RIPK1 and RIPK3 in detergent-insoluble fractions of wild-type BMDMs varying kinetics (Figure 4—figure supplement 2A-C). TRIF maximally accumulated by approximately 1 hour (Figure 4—figure supplement 1A), whereas maximal levels of RIPK1 and RIPK3 were observed at approximately 6 hours of treatment (Figure 4—figure supplement 1B, C). This is consistent with the kinetics observed in *Atg16l1*-cKO BMDMs over the same time course (Figure 4B, TRIF; Figure 4C, RIPK1; Figure 4D, RIPK3). In contrast, basal turnover of monomeric TRIF, RIPK1 and RIPK3 was not significantly affected by lysosomal inhibition (Figure 4—figure supplement 2D-F). Proteasomal inhibition with MG132 provided modest and transient accumulation of TRIF or RIPK1 (Figure 4—figure supplement 2D, TRIF; Figure 4—figure supplement 2E, RIPK1), while basal RIPK3 levels were not affected (Figure 4—figure supplement 2F). Thus, we conclude that autophagy specifically regulates the lysosomal turnover of active forms of TRIF, RIPK1 and RIPK3. These new data are discussed in the subsection “Loss of autophagy results in accumulation of active forms of TRIF, RIPK1 and RIPK3 during necroptosis.

- As we highlighted before, the technique used to induce necroptosis is well established in the field. However, it is known that this method induces autophagy. Would it be possible to use another technique to induce necroptosis, without targeting autophagy?

This is challenging to address, since autophagy is an acknowledged stress response pathway induced under numerous inflammatory states. We have utilized multiple TLR and cytokine (TNF) ligands to induce necroptosis and arrived at a consistent conclusion that autophagy/ATG16L1 is cytoprotective under all states, thus we expect autophagy to be broadly induced during necroptotic innate immune response in macrophages. Our data support a framework where autophagic flux promotes the degradation of RHIM-domain proteins under multiple contexts, thus it is likely that most stimuli promoting the aggregation of TRIF, RIPK1 or RIPK3 depend on autophagy to curb necroptosis. We have revised our schematic to depict the role of autophagy in suppressing TLR and cytokine-mediated necroptosis (Figure 7—figure supplement 1B, C), and hope this provides a satisfactory description of the role of autophagy in necroptosis.

- Figure 3 shows accumulation of detergent-insoluble forms of TRIF, RIPK1 and RIPK3. Authors comment in the Discussion (second paragraph) about the possibility of protein ubiquitination playing a role in this process. We consider that detecting ubiquitination of these proteins will support their hypothesis considering that authors make a strong statement about the role of selective autophagy in this process.

Demonstrating ubiquitination of active forms of RIPK1, RIPK3 and TRIF in primary macrophages is a technically challenging request, given the requirement of highly denaturing lysis conditions. Nonetheless, we have attempted to measure ubiquitination of TRIF, RIPK1 and RIPK3 in primary BMDMs under these conditions. Using a mixture of M1 and K63-linkage-specific antibodies, we successfully immunoprecipitated autophosphorylated RIPK1 and RIPK3 from wild-type and *Atg16l1*-cKO BMDMs following LPS/zVAD treatment. As shown in Figure 4E, we demonstrate that loss of *Atg16l1* results in accumulation of poly-ubiquitinated forms of autophosphorylated RIPK1 and RIPK3 upon LPS/zVAD treatment. This is entirely consistent with our previous observations that high molecular weight forms of autophosphorylated RIPK1 and RIPK3 accumulate in *Atg16l1*-cKO BMDMs (Figure 3D, F; Figure 4C, D).

Unfortunately, we have been unable to obtain publication-quality immunoblots demonstrating the accumulation of ubiquitinated TRIF in the same experimental setting. While we consistently observe high MW ubiquitinated TRIF in *Atg16L1-*cKO BMDM lysates, a clear demonstration is precluded by a considerable non-specific signal, perhaps from degradation of primary antibody (Author response image 1). Thus, we have focused our discussion on RIPK1 and RIPK3 ubiquitination in the revised manuscript. We hope this effort provides sufficient evidence to demonstrate that active RIPK1 and RIPK3 are indeed ubiquitinated during necroptosis, and that defective autophagy enhances the accumulation of ubiquitinated RIPK1 and RIPK3. These data are discussed in the second paragraph of the subsection “Loss of autophagy results in accumulation of active forms of TRIF, RIPK1 and RIPK3 during necroptosis”.

**Author response image 1. respfig1:** TRIF, autophosphorylated RIPK1 and autophosphorylated RIPK3 are ubiquitinated during necroptosis. Immunoblots of TRIF, autophosphorylated RIPK1, autophosphorylated RIPK3 and ubiquitin in BMDM lysates following immunoprecipitation of M1 or K63-ubiquitinated proteins after 4 hours of LPS/zVAD treatment. Red arrow depicts specific high MW TRIF signal.

Reviewer #2:This manuscript extends on observations reported by this laboratory in Samie et al., 2018, which reported Tax1BP1 mediated selective autophagy of TRIF. Here, the authors demonstrate that oligomerised RHIM domain proteins (including TRIF) accumulate in ATG16L1 deficient macrophages. They investigate autophagy as a possible homeostatic mechanism for clearing these oligomers, thereby opposing inflammation and cell death. Consistent with this, loss of essential autophagy genes resulted in enhanced cell death in BMDMs stimulated with LPS or Poly I:C and zVAD. A notable strength of this paper which distinguishes it from previous observations on RIP kinases and autophagy is that the authors provide evidence of involvement of the FIP200 and ATG14, which indicates this is due to canonical macroautophagy and not a 'non-canonical' or LAP-like process. The loss of Rubicon makes no difference, and whilst this does not rule out a contribution from 'non-canonical' autophagy (see discussion in Fletcher, Ulferts et al. EMBOJ 2018), taken together with the other genetic data it strongly argues against LAP being required here. The data to support this are shown in Figures 1F and Supplementary Figure 2B, the paper would be much easier to read if these data were grouped together.

These datasets are now grouped together as Figure 1D and E in the revised manuscript.

The authors demonstrate that Necrostatin fails to completely suppress necroptosis in autophagy deficient cells, and show this depends on Mlkl and Ripk3. ATG16L1 deficient cells accumulate detergent insoluble forms of RHIM domain proteins, consistent with a role for autophagy in their degradation. The authors go on to show that ZBP1 accumulates in ATG16L1 deficient cells, but that curiously this protects against necroptosis. This is interesting, but the effect only seems to take place in the setting of ATG16L1 deficiency so it remains unclear if there is a physiological relevance.Absent from this paper are any direct measurements of autophagy or autophagic flux. It would be very helpful to the authors' argument to demonstrate that oligomerised RHIM-domain proteins do indeed become selectively engulfed by autophagosomes. As a minimum I would expect to see data on accumulation of autophagosomes and associated lipidation of LC3 after RHIM-domain oligomerisation, and (crucially) the effect of ATG14/FIP200/ATG16L1 deletion on this process. This would provide additional strong evidence of regulation of signalling by canonical autophagy.

We thank the reviewer for suggesting a more direct measurement of autophagic flux. While deletion of *Atg16l1* revealed the accumulation of RHIM-domain proteins in macrophages, we acknowledge that it is not possible to study autophagic flux in this genotype. Thus, we treated wild-type BMDMs with the lysosomal inhibitor Bafilomycin A1 during a kinetic analysis of necroptosis. This is an established method to investigate the process of autophagic degradation of cytosolic cargo, as it blocks the lysosomal degradation of putative autophagy substrates, forcing their accumulation in the lysosomal compartment. Consistent with our observations in *Atg16l1*-cKO BMDMs, blocking lysosomal degradation in wild-type BMDMs resulted in the accumulation of TRIF, RIPK1 and RIPK3 in detergent-insoluble compartments over 6 hours of LPS/zVAD treatment (Figure 4—figure supplement 2A-C). Importantly, LC3B lipidation was rapidly induced (at approximately 30min) upon LPS/zVAD treatment, as observed by accumulation of LC3-II in Bafilomycin A1 treated BMDMs (Figure 4—figure supplement 2G). Finally, we measured the accumulation of multiple autophagy receptors under the same conditions and observed their accumulation in detergent-insoluble fractions (Figure 4—figure supplement 2H). Together, these data provide evidence that autophagic flux is induced during necroptosis, and that active forms of TRIF, RIPK1, RIPK3 are indeed trafficked to the lysosomal compartment for degradation. Comparing kinetics of RHIM-protein aggregation between wild-type and *Atg16l1*-cKO BMDMs reveals a consistent kinetic, with TRIF accumulation preceding that of autophosphorylated as well as total RIPK1 and RIPK3. In all these settings, loss of *Atg16l1* enhanced the accumulation of RHIM-domain proteins (Figure 4B-D). Additionally, we were able to assay ubiquitinated forms of active (autophosphorylated) RIPK1 and RIPK3 and demonstrate increased accumulation in ATG16L1 deficient BMDMs during necroptosis (Figure 4E). These data are described in the second paragraph of the subsection “Loss of autophagy results in accumulation of active forms of TRIF, RIPK1 and RIPK3 during necroptosis”.

I accept the authors argument that a full mechanistic explanation of exactly how the proteins are ubiquitinated and degraded (assuming this is the case) is beyond the scope of the paper, but given their previous findings of Tax1BP1 being a critical component required for degradation of TRIF (at least in mice, the situation in humans may well be different), it is surprising that the authors do not comment at all on this. It would greatly strengthen the paper to see data on the requirement (or otherwise) of autophagy receptors such as TaxBP1, p62 and (in humans) NDP52.

We have successfully generated CRISPR-mediated knockdown of *Sqstm1/p62, Tax1bp1* or *Calcoco1* in wild-type BMDMs (Figure 5C). These autophagy receptors were previously identified as candidate receptors accumulating in *Atg16L1*-cKO BMDMs (Samie et al., 2018). Induction of LPS/zVAD or PolyI:C/zVAD-mediated necroptosis revealed that loss of Tax1bp1 significantly increased BMDM necroptosis (Figure 5D). Consistent with the phenotypes of BMDMs lacking ATG16L1, ATG14L or FIP200, TAX1BP1 deficiency resulted in Necrostatin-1 resistant cell death following activation of TRIF signaling via TLR3 or TLR4 (Figure 5D, PolyI:C+zVAD+Nec-1 or LPS+zVAD+Nec-1 conditions). These data are described in the subsection “The autophagy receptor TAX1BP1 prevents TRIF-mediated necroptosis”.

In summary this paper provides an important if somewhat limited advance in our understanding of the regulation of necroptosis. The conclusions are in general well supported by the data. The paper would be greatly strengthened by exploring the mechanisms of selective autophagy in this setting.Additional data files and statistical comments:

Many of the pairwise analyses throughout require correction for multiple hypotheses.

We have updated all pairwise analyses to correct for multiple hypotheses (Holms-Sidak method, α=0.05). This has not affected our conclusions.Reviewer #3:[…] Overall the paper is well written and the data very compelling. However some additional insights on how ZBP1 can repress necroptosis would strengthen this study.– The authors should provide some insights into how ZBP1 functions in repressing necroptosis.

This is a highly relevant comment but broad in its implications. Studies to address the molecular mechanism(s) by which ZBP1 functions in repressing necroptosis requires additional tools (e.g. domain-specific ZBP1 mutants) for primary macrophage complementation assays. Specifically, RHIM-mutant vs. Za-domain mutant versions of ZBP1 would need to be generated and knock-in cells obtained both in wild-type and autophagy-deficient backgrounds. These requirements place a thorough assessment of ZBP1 function outside the scope of the current manuscript. However, we have highlighted this possibility in the Discussion to encourage future investigation on the molecular mechanism by which ZBP1 represses TRIF-mediated necroptosis (Discussion, first paragraph).

– Ripk1 inhibition could not completely block necroptosis when TRIF is involved (TLR3 and TLR4 stimuli in autophagy deficient cells). Can RIPK3 be recruited to the TRIFosome and activated in the absence of RIPK1? Is it just the kinase activity of RIPK1 that becomes redundant (i.e. is scaffolding still required)? Use of RIPK1 sgRNA could address this.

Since RIPK1 deletion significantly compromises hematopoietic cell and macrophage fitness (Rickard, J.A., et al. Cell 2014; Roderick, J.E. et al. PNAS 2014; Newton, et al., 2016), assessment of TRIFfosome formation in RIPK1-deficient BMDMs requires the generation of *Atg16l1*-cKO; *Ripk1*-KO; *Caspase-8*-KO; *Ticam1*-KO quadruple knockout or cells. Additional deletion of *Mlkl* may be needed to maintain viability in the absence of *Ripk1* or *Caspase-8* if TLR3 or TLR4-mediated pathways are investigated. This complicated genetic background poses a significant technical challenge and is out of scope for the current study.

Previous work by the Mockarski lab (Kaiser et al., 2013) and Wang lab (He, S. et al. PNAS 2011) have shown that TRIF activation directly stimulates RIPK3 auto-phosphorylation and downstream necroptosis via MLKL. Kaiser et al. demonstrate that the TRIF/RIPK3 pathway is activated in either the absence of RIPK1 protein or its kinase activity, thus revealing a RIPK1-independent pathway of TRIF-mediated necroptosis. Our observations are consistent with these previously described mechanisms; defective autophagy likely enhances TRIF/RIPK3-mediated necroptosis in the absence of RIPK1. We raise this possibility in the Discussion (first paragraph) and have updated the references to cite the above works in the revised manuscript.

– How is necroptosis triggered in non-TRIF mediated TLR triggered necroptosis? Through autocrine secretion of TNF or IFN?

To address this query, we blocked TNF- or IFN-signaling by pre-treating BMDMs with TNFR2-Fc or anti-IFNAR1, respectively. Consistent with previous studies, we observed that non-TRIF mediated necroptosis of wild-type BMDMs was rescued by blockade of TNF or IFNAR1 (Figure 2—figure supplement 2A, E, F). However, this failed to rescue necroptosis of *Atg16l1*-cKO BMDMs to the same level as Necrostatin-1 (Figure 2—figure supplement 2A, D, E). These findings reveal that RIPK1 activity dominates the necroptotic phenotype when autophagy is defective and non-TRIF mediated TLR signaling engaged. This is likely due to additional death ligands contributing to RIPK1 activity. We discuss these new findings in the second paragraph of the subsection “TRIF and RIPK1 drive necroptosis in *Atg16l1*-deficient macrophages” and provide new data (Figure 2—figure supplement 2A-F) in the revised manuscript.

– The recent paper (Sarhan et al., 2018) suggests that autocrine IFN is required for necroptosis. Is IFN involved in necroptosis when autophagy is not functional? Does perturbed clearance of TRIF and RIPK1 result in higher IFN signaling and thus higher cell death?

We thank the reviewer for this query. To assay IFN signaling, we measured STAT1 phosphorylation in wild-type and *Atg16l1*-cKO BMDMs over a 6-hour time-course of necroptosis induced by LPS/zVAD. Loss of *Atg16l1* enhanced STAT1 phosphorylation, both in magnitude and kinetics (Figure 2—figure supplement 2G). Pharmacological blockade of IFNAR1 rescued necroptosis of wild-type BMDMs, consistent with Sarhan et al., 2018. Interestingly, it also decreased necroptosis of *Atg16l1*-cKO BMDMs comparably to Necrostatin-1, indicating that elevated IFN signaling contributes to elevated cell death (Figure 2—figure supplement 2B, C). These data provide an additional insight into autocrine signaling that precipitates inflammation when autophagy is perturbed. We present and discuss these new findings in the third paragraph of the subsection “TRIF and RIPK1 drive necroptosis in Atg16l1-deficient macrophages”.

– Figure 3D and F suggests that when autophagy is deranged, RIPK1- and RIPK3- oligomerization is dependent on TRIF but not phosphorylation of RIPK1 and RIPK3?

When autophagy is defective (e.g. *Atg16l1* deletion), TRIF-mediated TLR stimulation elevates autophosphorylation as well as oligomerization of RIPK1 and RIPK3. Knockdown of TRIF in ATG16L1-deficient BMDMs decreased autophosphorylation of both high MW and monomeric RIPK1 (Figure 3D). Levels of autophosphorylated, high MW RIPK3 were more significantly affected, with a considerable loss of this form of RIPK3 when TRIF is deleted. These findings are consistent with the previously discussed observations that TRIF can directly engage RIPK3 and drive its signaling (He, S. 2011; Kaiser, 2013). We discuss direct RIPK3 engagement by TRIF and the impact of defective autophagy in the first paragraph of the Discussion.

– The term "screening" is a bit misleading in Figure 1. They optimized electroporation conditions for delivery of sgRNA:Cas9 RNPs into primary BMDMs. This could be added to the supplementary figure.

We have replaced “screening” with “comparing” to address this query. We have also moved the schematic to Figure 1—figure supplement 2A, B in the revised manuscript.